# History and dynamics of Fennoscandian Ice Sheet retreat, contemporary ice-dammed lake evolution, and faulting in the Torneträsk area, northwestern Sweden

Karlijn Ploeg[1,2,3] and Arjen P. Stroeven[1,2]

[1]Department of Physical Geography, Stockholm University, Stockholm, Sweden
[2]Bolin Centre for Climate Research, Stockholm University, Stockholm, Sweden
[3]Now at: Department of Earth Science, University of Bergen, Bergen, Norway

**Correspondence:** Karlijn Ploeg (karlijnploeg@gmail.com)

**Abstract.**

The prospect of alarming levels of future sea level rise in response to the melting of the Antarctic and Greenland ice sheets affirms an urgency to better understand the dynamics of these retreating ice sheets. The history and dynamics of the ephemeral ice sheets of the Northern Hemisphere, such as the Fennoscandian Ice Sheet, reconstructed from glacial geomorphology, can thus serve as a useful analogue. The recent release of a 1 m LiDAR-derived national elevation model reveals an unprecedented record of the glacial geomorphology in Sweden. This study aims to offer new insights and precision regarding ice retreat in the Torneträsk region of northwestern Sweden, and the influence of ice-dammed lakes and faulting on the dynamics of the ice sheet margin during deglaciation. Using an inversion model, mapped glacial landforms are ordered in swarms representing spatially and temporally coherent ice sheet flow systems. Ice-dammed lake traces such as raised shorelines, perched deltas, spillways, and outlet channels, are particularly useful for pin-pointing precise locations of ice margins. A strong topographic control on retreat patterns is evident, from ice sheet disintegration into separate lobes in the mountains to orderly retreat in low-relief areas. Eight ice-dammed lake stages are outlined for the Torneträsk Basin, the lowest of which yields lake extents more extensive than previously identified. The three youngest stages released a total of 26 $km^3$ of meltwater as glacial lake outburst floods (GLOFs) through Tornedalen, changing the valley morphology and depositing thick deltaic sequences in Ancylus Lake at its highest postglacial shoreline at around 10 cal ka BP. The Pärvie Fault, the longest-known glacially-induced fault in Sweden, offsets the six oldest lake stages in the Torneträsk Basin. Cross-cutting relationships between glacial landforms and fault scarp segments are indicative of the Pärvie Fault rupturing multiple times during the last deglaciation. Precise dating of the two bracketing raised shorelines or the ages of the corresponding GLOF sediments would pinpoint the age of this rupture of the Pärvie Fault. Collectively, this study provides data for better understanding the history and dynamics of the Fennoscandian Ice Sheet during final retreat, such as interactions with ice-dammed lakes and re-activation of faults through glacially-induced stress.

# 1 Introduction

Anthropogenic climate warming has caused a quantifiable reduction in global ice volume since pre-industrial times (IPCC, 2023). This melting of glaciers, ice caps, and ice sheets constitutes a significant contribution to global mean sea level rise (Slater et al., 2020; Box et al., 2022), which is already posing risks to vulnerable low-elevation coastal communities through flooding, saltwater intrusion, and coastal erosion (Nicholls and Cazenave, 2010; Mentaschi et al., 2018; Taherkhani et al., 2020; Levy et al., 2024). The impact of this meltwater release also impacts other components of the Earth system such as ocean mixing and atmospheric circulation (Golledge et al., 2019; Li et al., 2023).

The urgency of acquiring more knowledge on ice sheet dynamics, particularly on spatial and temporal responses to future warming, is emphasized by the likelihood of accelerated ice sheet melt in the coming centuries (Briner et al., 2020; van de Wal et al., 2022). However, research into, and monitoring of, the Greenland Ice Sheet and the Antarctic Ice Sheet have been curtailed by a relatively brief period of observation (Mouginot et al., 2019; Rignot et al., 2019; Hanna et al., 2024). Hence, direct information on ice sheet evolution often pertains to short-term, near-margin, studies (e.g., Bentley et al., 2014; Groh et al., 2014) while future predictions rely on such data to constrain numerical ice sheet models (Stokes et al., 2015; Ely et al., 2021; Suganuma et al., 2022; Coulon et al., 2024). An alternative to this data-starved approach is to glean information on ice sheet behavior from data-rich environments of the formerly-glaciated landscapes in the Northern Hemisphere.

Large ice sheets have repeatedly covered the Northern Hemisphere, particularly during the last 2.6 million years (Kleman et al., 2010; Hughes et al., 2016; Batchelor et al., 2019). Because the North American Ice Sheet and the Eurasian Ice Sheet complexes during their last maximum extents, and particularly their Laurentide Ice Sheet and Fennoscandian Ice Sheet (FIS) components, respectively, had configurations similar to the Greenland and Antarctic ice sheets today, they can serve as credible analogues to understand ice sheet response to climate change. Understanding this response can be achieved through a reconstruction of ice sheet extent and dynamics during the last deglaciation using the geomorphological record (Kleman and Borgström, 1996; Kleman et al., 2006; Stroeven et al., 2016, 2021; Chandler et al., 2018; Greenwood et al., 2024; Szuman et al., 2024).

The FIS was the largest sector of the Eurasian Ice Sheet complex during the last glaciation, which, at its maximum extent, merged with the British-Irish Ice Sheet and the Svalbard-Barents-Kara Ice Sheet (Hughes et al., 2016). Eurasian Ice Sheet dynamics (Patton et al., 2016, 2022) and disintegration (Hughes et al., 2016; Patton et al., 2017) were largely initiated along its extensive marine margin. The most recent reconstruction of FIS deglaciation by Stroeven et al. (2016) is based on geomorphological and geochronological data, and it refines and extends several earlier reconstructions (Lundqvist, 1986; Lundqvist and Saarnisto, 1995; Kleman et al., 1997; Boulton et al., 2001). These reconstructions precede the advent of LiDAR data and address ice retreat on a continental scale. Hence, they do not realistically address deglaciation patterns in the mountains due to the spatial complexity of the geomorphological evidence in mountainous terrain (Kleman, 1992). Hence, ice sheet dynamics across topographically challenging terrain, as has been shown for the Cordilleran Ice Sheet (e.g., Seguinot et al., 2016; Dulfer et al., 2022), remains an important research gap in Scandinavia.

In this study, the deglaciation dynamics of the FIS are revisited for the Torneträsk region in northern Sweden. The Torneträsk region (Fig. 1a) has been one of Sweden's premier sites for studying geomorphology, with a particular focus on the last deglaciation (Melander, 1980; Stroeven et al., 2002) and the formation of series of ice-dammed lakes during ice sheet retreat (e.g., Sjögren, 1908, 1909; Holdar, 1952, 1957, 1959; Melander, 1977c, 1980). Using a new (2021) LiDAR-based elevation model provided by Lantmäteriet, the Swedish Mapping, Cadastral, and Land Registration Authority, landforms are mapped that detail the retreat history of the FIS. The LiDAR-data circumvents problems inherent in the use of aerial photographs (e.g., forest cover), on which the most recent detailed geomorphological maps of the Torneträsk region were based, albeit along with extensive field verification (Melander, 1977a, b). Refining ice-dammed lake reconstructions impacts the precision of reconstructed patterns of ice sheet retreat (e.g., Jansson, 2003; Utting and Atkinson, 2019; Regnéll et al., 2019, 2023; Dulfer et al., 2022; Romundset et al., 2023), which is especially valuable as the dynamics of ice sheet demise in topographically challenging terrain remains understudied in Scandinavia (Borgström, 1989; Kleman et al., 2020; Regnéll et al., 2019, 2023; Romundset et al., 2023). The aim of this study is therefore to refine the reconstruction of the deglaciation of the Torneträsk region by improving the reconstruction of its ice-dammed lake systems.

The former existence of ice-dammed lakes has allowed for detailed regional reconstructions of ice sheet retreat stages (e.g., Lundqvist, 1972; Jansson, 2003; Jakobsson et al., 2007; Perkins and Brennand, 2015; Høgaas and Longva, 2018; Utting and Atkinson, 2019; Regnéll et al., 2019, 2023). Indeed, ice-dammed lakes, dammed between the Scandinavian mountain range water divide and the retreating FIS margin outline the location of its terminal configuration (e.g., Svenonius, 1898; Lundqvist, 1972; Stroeven et al., 2016; Regnéll et al., 2019, 2023). Additionally, the distribution and longevity of ice-dammed lakes shed light on their interactions with ice sheet margins, of which knowledge is slowly mounting (Utting and Atkinson, 2019; Carrivick et al., 2020; Mallalieu et al., 2021; Scherrenberg et al., 2023; Zhang et al., 2024). Finally, a refined history of ice-marginal retreat potentially enables future investigations of the interaction between the changing configuration of the retreating ice sheet, its marginal positions, and a re-activation of faults through the overprinting of the prevailing regional tectonic stress with glacially-induced stress (Fig. 1b).

## 2 Study area

The study area is located in northernmost Sweden (Fig. 1a). The research is centered around the WNW-ESE trending valleys Torneträsk and Rautasjaure (Fig. 1c). The Torneträsk Basin cuts through the Scandinavian mountain range, also known as the Scandes, and drains to the east. Across the border to Norway, over a pass to the west, are the headwaters of Rombaksfjorden (Rombaken; Fig. 2a). The mountain range trends along the long axis of the Scandinavian Peninsula and straddles the border between Sweden and Norway. The study area is approximately 5980 $km^2$ and the irregular shape of its western margin follows the outline of the international border.

Based on relief, the study area can be divided into a montane region in the west and a premontane region in the east (Fig. 1c). The montane region has several peaks above 1000 m in elevation, with the highest peak Kåtotjåkka at 1986 m above sea level (a.s.l.). The landscape has experienced extensive glacial erosion, as demonstrated by deep glacial valleys (e.g., as seen

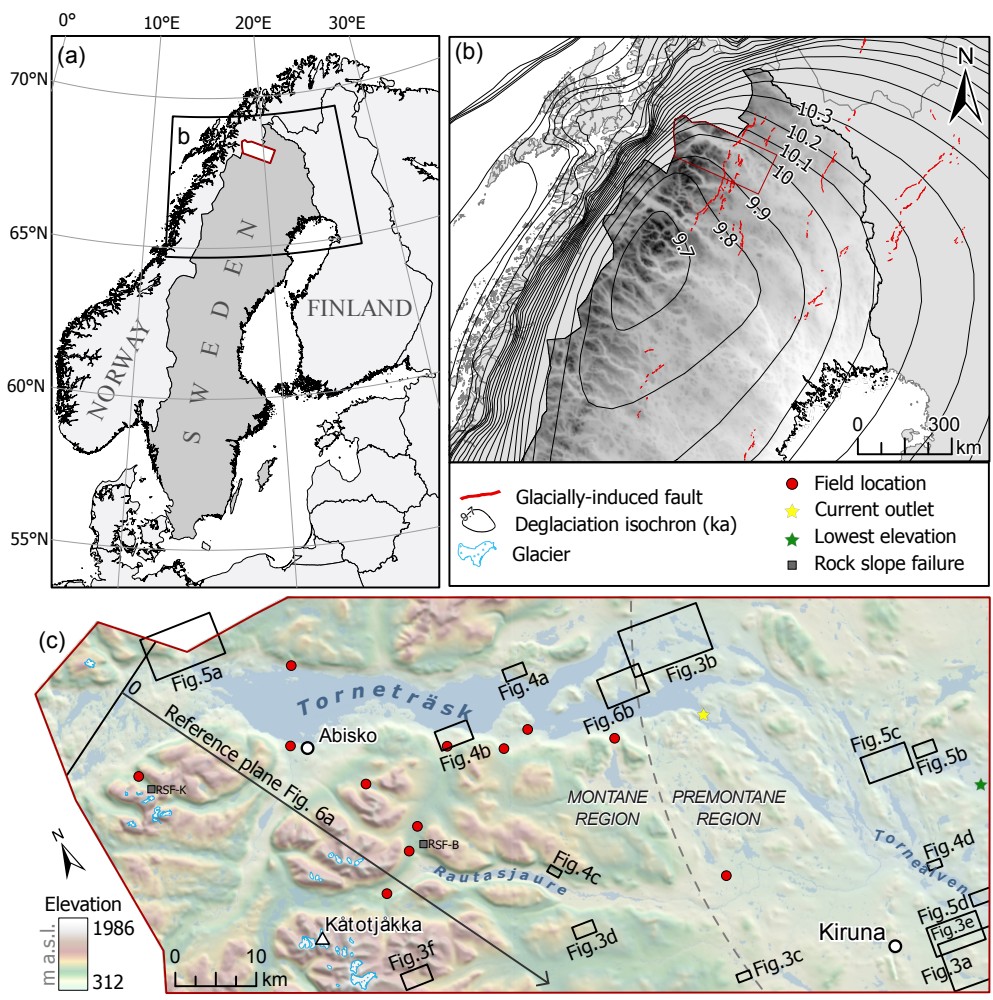

**Figure 1. Overview of the study area.** Background in (b) and (c) is the National Elevation Model by ©Lantmäteriet. (a) Outline of the study area in Sweden. (b) Deglaciation isochrons (cal ka BP) in northern Sweden (Stroeven et al., 2016) and glacially-induced faults (Munier et al., 2020). (c) Study area of the Torneträsk region, including the border between the premontane and montane regions based on Hättestrand (1998). Black boxes indicate the position of maps in Figs. 3a–3f, 4a–4d, 5a–5d, and 6b.

in the irregular border between Caledonian Nappes and the Precambrian basement rocks it covers; Ebert et al., 2011), cirques, and scoured bedrock (Kleman and Stroeven, 1997; Jansson et al., 2011). Glacially-scoured bedrock outcrops are especially
abundant across the border within the Torneträsk depression (Stroeven et al., 2002). In contrast, significant upland areas have escaped glacial erosion and they are typically characterized by gentle slopes, round summits with occasional tors, wide shallow fluvial valleys, and open passes (Kleman and Stroeven, 1997). The premontane region is a low relief area ranging between 300 and 600 m a.s.l., which consists of extensive plains with residual hills (Lidmar-Bergström, 1995; Ebert and Hättestrand, 2010). The lowest elevation in the study area at 312 m a.s.l. is found along Sevujoki, draining lake Sevujärvi towards the southeast

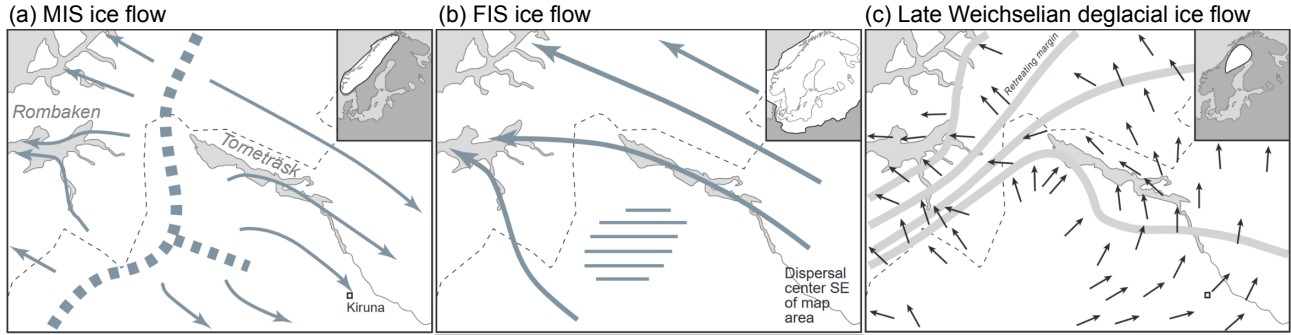

**Figure 2. Ice flow history of the Torneträsk Basin and Rombaken.** (a) Ice flow during mountain ice sheet (MIS)-style ice configurations with an ice divide over the mountains. (b) Ice flow during FIS-style ice configurations with an ice divide over the Gulf of Bothnia. (c) Ice flow during the last deglaciation with a final ice divide southwest of Torneträsk Basin. Modified from Stroeven et al. (2002).

(Fig. 1c). The region is characterized by long and narrow lakes formed by glacial deepening and widening of preglacial river valleys (Hättestrand, 1998).

The area is known for its palimpsest landform systems (Kleman, 1992; Kleman and Stroeven, 1997). The existence of relict areas is attributed to the patchy erosional impact of the overriding FIS, reflecting a spatially-varying basal thermal regime (Stroeven et al., 2002, 2013; Goodfellow et al., 2008). Glacial deposits and geomorphology indicate there were two modes of ice sheet configuration during the Quaternary: elongated west-centered mountain ice sheets (MIS), and large east-centered ice sheets (various configurations of FIS) (Kleman and Stroeven, 1997; Kleman et al., 2008). During MIS-style glaciations ice flow was directed eastward along the Torneträsk Basin (Fig. 2a), while during FIS-style glaciation ice flow was directed westward, where Torneträsk valley was probably one of the largest outlets of the FIS (Fig. 2b, Stroeven et al., 2002). Deglacial flow is characterized by a shift from western ice flow, to northern ice flow, to northeastern ice flow (Fig. 2c, Stroeven et al., 2002, 2016).

Torneträsk is the largest lake within the study area with a surface area of 332 km$^2$. The largest inlet of Torneträsk, river Abiskojåkka, enters the lake immediately west of Abisko village (Fig. 1c). The current outlet is at Tarrakoski fors (Fig. 1c), which is a set of rapids that drains Torneträsk to the 3 m lower water level of lake Alajärvi, which eventually connects to Torneälven in Tornedalen (Fig. 1c). The current lake level of Torneträsk is at 342 m a.s.l., although the lake level varies over a meter throughout the year (SMHI, 2020). The lake has a maximum depth of 168 m according to bathymetric measurements from 1920/1921 (SMHI, 2020), but recent sonar measurements by Abisko Research Station (in Abisko) show that its depth, in places, exceeds 190 m (A. Kristoffersson, personal communication, 26 January 2022).

## 3  Methods

### 3.1  Data sets

Several remotely-sensed raster data sets and processed versions were used for the identification and mapping of landforms. Lantmäteriet provides a LiDAR-based digital elevation model (DEM), also known as the National Elevation Model, with a spatial resolution of 1 m (Lantmäteriet, 2021b). The DEM has an absolute vertical accuracy of <0.1 m and an absolute horizontal accuracy of <0.3 m, but these can vary depending on point density of the laser scanning, time of scanning, and survey technique (Lantmäteriet, 2021b). The DEM was processed in ArcGIS Pro 2.9.3 to create a hillshade relief model using an illumination angle with an altitude of 30° and azimuths of 45° and 315°, as these are considered the optimal values for the visualization of hillshade relief models for the purpose of glacial geomorphological mapping (Smith and Clark, 2005; Hughes et al., 2010). Additional azimuths of 90° and 180°, perpendicular and parallel to the dominant lineation orientation, respectively, were applied to reduce the 'azimuth bias' (Smith and Clark, 2005; Chandler et al., 2018). Additionally, a slope model was derived from the DEM and contour lines were created with intervals of 10 m, 20 m, and 100 m. Lantmäteriet (2021a) provides natural colour (RGB 4, 3, 2) and colour-infrared (RGB 5, 4, 3) orthophotos with a resolution of 0.5 m acquired in 2018 and 2021 for roughly the western montane region and eastern premontane region, respectively (Fig. 1c). Although the mapping was primarily DEM-based, in certain cases the aerial imagery enhanced landform detectability. Google Earth Pro enabled visualization of the terrain in 3D, which was mostly used to cross-check mapping based on other imagery. Its imagery is primarily satellite-based and displays varying resolutions.

Vector data sets of previously published studies were used for different purposes. The international database of Munier et al. (2020) contains glacially-induced faults in northern Fennoscandia (Fig. 1b), of which many were previously proposed and recently confirmed based on the recent LiDAR data. The faults in the database were cross-referenced with the LiDAR-based DEM, but no effort was made to identify new faults. The data set was used to identify cross-cutting relationships between glacial landforms and fault scarps. The deglaciation isochrons reconstructed by Stroeven et al. (2016) were used to evaluate the implications of the direction of mapped landforms and to constrain the chronology (Fig. 1b). Cosmogenic nuclide [10]Be exposure ages of two rock slope failure (RSF) deposits were taken from Stroeven et al. (2002, 2016). The RSF extents were cross-referenced against the LiDAR-based DEM. The printed geomorphological maps by Melander (1977a, b) and Hättestrand (1998) were digitized and georeferenced in GIS software using locations on the map with known coordinates for cross-referencing purposes.

### 3.2  Fieldwork

Fieldwork was conducted in August 2021. The aim of the fieldwork was to ground-truth landforms. At every site, landform and landscape morphology and sedimentological properties of the landform were assessed. The field sites are concentrated around the southern shore of lake Torneträsk (Fig. 1c), given the focus on ice-dammed lake traces, such as raised shorelines and perched deltas, and ease of access.

## 3.3 Mapping and analysis

Remote mapping on the scale of a regional sector of a paleo-ice sheet requires a systematic mapping approach, in order to map a large area in a time-efficient, yet accurate manner (Chandler et al., 2018). Relevant for the mapping process itself is the construction of a landform identification table, which describes the morphology, dimensions, identification criteria, and paleoglaciological significance of each glacial landform (Table 1). Landforms were mapped at a scale of 1:30 000, although the final map is presented at a scale of 1:300 000 (Fig. S1). Depending on the landform, zooming in to a larger scale was required to outline the landforms correctly. Mapping was performed in two iterations.

The application of an inversion model is required for extracting ice-sheet properties from mapped glacial geomorphology, such as its thermal regime, subglacial hydrology, or the presence of ice streams. Here, the conceptual framework of Kleman et al. (2006) is applied to deduce ice sheet evolution through time. The inversion model is composed of a set of assumptions (Kleman et al., 2006, p. 196), a classification system for glacial landform assemblages, and a procedure for managing the landform data and incorporating absolute chronological data. The model thus explains how individual landforms are interpreted in terms of ice sheet properties, which results in ice sheet-wide glaciologically-consistent patterns by aggregation of the individual landforms into swarms. Wet-bed deglaciation swarms include eskers with aligned lineations. These fields of lineations and eskers are formed time-transgressively, parallel to ice flow, and perpendicular to the ice margin. Dry-bed deglaciation swarms typically lack subglacial landforms, due to an absence of sliding when the ice sheet is frozen to its substrate, but include meltwater channels, ice-dammed lake shorelines, and perched deltas. Such meltwater traces are imprinted on a relict surface, which can be non-glacial or glacial, thus demonstrating the subglacial preservation of landforms and landscapes. Ribbed moraine forms when subglacial conditions change from dry to wet-bed (Hättestrand and Kleman, 1999), with its individual ridges oriented perpendicular to ice flow direction. A set of these landforms representing coherent ice flow directions and ice margins can then be outlined to realistically visualize retreat patterns.

The mapping approach, that is, how the landforms are delimited in GIS software, is briefly described for all landforms in Table 1. Given the focus on ice-dammed lake traces, the mapping approach of raised shorelines and perched deltas, and the methodology to identify ice-dammed lake stages, are described in more detail below. Raised shorelines are mapped as polylines positioned midway between the toe and inner break of the shoreline. The polylines are then converted to vertices, for which the elevation in meters above sea level was extracted from the DEM by means of the *Add Surface Information tool* in ArcGIS Pro. Similarly, polylines of identified lake outlet channels are converted to vertices, for which the elevation was extracted for the first vertex, representing the minimum threshold of overflow. Perched deltas are mapped as polygons that demarcate the flat top surface. The minimum elevation along the polygon was extracted in the assumption that it represents the delta front. The elevations of the shorelines, perched deltas and outlet channels are analysed against the distance along a reference plane, of which the distance was calculated by means of the *Near tool* in ArcGIS Pro. The reference plane has an orientation of 325°N (Fig. 1c), which is oriented perpendicular to the isobases of raised shorelines as inferred from literature (Møller, 1987; Regnéll et al., 2019). The tilting of the shorelines is described as a gradient in $\mathrm{mkm^{-1}}$, where the elevation difference (in meters) is given over the distance (in kilometers) in the direction of the reference plane. The shorelines corresponding to specific lake

**Table 1.** Landform classification table describing the morphology, dimensions, possible identification errors, paleoglaciological significance, and the mapping approach of the landforms mapped in this study.

| Landform | Morphology | Dimensions | Possible identification error | Paleoglaciological significance | Mapping approach | Literature |
|---|---|---|---|---|---|---|
| **SUBGLACIAL** | | | | | | |
| **Lineation** | Elongated ridges, both depositional and erosional. Tend to occur in swarms. **(Figs. 3a–c, e)** | Meters long and tens of centimeters high to kilometers long and tens of meters high | May be confused with bedrock structures, although hillshades with multiple illumination angles may clarify | Formed parallel to ice flow, landform asymmetry reflects ice flow direction, reflects deglacial ice flow when occurring together with eskers | Polyline along crest | Clark et al. (2009); Benn and Evans (2010); Stroeven et al. (2016) |
| **Esker** | Single ridges or networks of parallel ridges. Typically sharp-crested, long and winding. **(Figs. 3b–c)** | Size up to hundreds of kilometers long and tens of meters high | Misinterpretation as type of moraine, although esker is usually more sinuous | Formed in subglacial tunnels parallel to ice flow and close to a retreating ice margin | Polyline along crest | Brennand (2000); Storrar et al. (2014); Livingstone et al. (2020); Stroeven et al. (2021) |
| **Subglacial meltwater channel** | Channels incised into bedrock or sediment, oriented oblique to slope. May connect to esker segments. **(Fig. 3c)** | Highly variable dimensions, up to several hundred of meters long | May be confused with submarginal lateral meltwater channels | Reflect ice sheet flow direction close to the ice margin | Polyline along thalweg of channel, arrow pointing downslope | Kleman (1992); Greenwood et al. (2007); Margold et al. (2013) |
| **Ribbed moraine** | Fields of curved ridges, regularly and closely spaced | Hundreds of meters long and tens of meters high | May be confused with solifluction lobes, although ribbed moraine are usually found in depressions | Formed transverse to ice flow, with outer limbs pointing down-ice. Indicative of a change from dry-bed to wet-bed conditions | Polygon demarcating fields of curved ridges | Hättestrand (1997); Hättestrand and Kleman (1999); Dunlop and Clark (2006); Benn and Evans (2010) |
| **ICE-MARGINAL** | | | | | | |
| **Moraine** | Straight or arcuate ridges, can occur in series. Potentially continuous, but often interrupted by gaps of non-deposition or erosion. **(Fig. 3f)** | From few meters to kilometers long and meters to tens of meters high | Can be difficult to distinguish from a protalus rampart in mountainous areas | Marginal moraines outline the shape and position of a former ice margin. Undifferentiated moraines are smaller equifinal landforms formed by different (ice-marginal) processes | Polyline along crest | Heyman and Hättestrand (2006); Benn and Evans (2010) |
| **Veiki moraine** | Semi-circular plateaus with a ridge along their rims. Whereas the rim is dry and covered by forest vegetation, the depressions within often consist of lakes or mires. | The plateaus cover areas from 0.1–30 $km^2$ and have a relief from 2–60 m | Possible to confuse with hummocky moraine if poorly developed | Formed through down wasting of debris-covered stagnant ice with ice-walled lakes on top | Polygon demarcating areas with multiple plateaus | Lagerbäck (1988); Hättestrand (2007); Clayton et al. (2008); Benn and Evans (2010); Alexanderson et al. (2022) |
| **Lateral meltwater channel** | Straight or winding channels cut into valley walls, subparallel to the contours. Often occurs in series. **(Fig. 3d)** | Tens of meters deep, hundreds of meters long, meters wide | Misinterpreting as bedrock structures, steplike solifluction lobes or shorelines, although the latter is strictly horizontal | Formation along ice margin, possible to infer ice surface slope and ice thickness | Polyline along thalweg of channel, arrow pointing downslope | Greenwood et al. (2007); Margold et al. (2013); Stroeven et al. (2016) |
| **PROGLACIAL** | | | | | | |
| **Proglacial meltwater channel** | Channel incised into bedrock or sediment, aligned to the local bed slope. **(Fig. 3e)** | Tens to over hundreds of meters long to tens of meters wide | May be confused with contemporary river incisions, but identifiable by dry-bed or underfitted stream. May be misinterpreted as outlet channel | Formation at terminus of the ice margin | Polyline along thalweg of channel, arrow pointing downslope | Greenwood et al. (2007); Benn and Evans (2010); Stroeven et al. (2021) |
| **Raised shoreline** | Zone of (nearly) horizontal wave washed till, eroded rock terrace or an accumulation of sediment. Gradient of 0.5 m km$^{-1}$. Characterized by a break in slope. Often occurs in series. **(Figs. 4a–d)** | Few meters wide, can extend hundreds of meters in length | Misinterpreting as bedrock structures (although those usually have a more extensive spatial distribution and are seldom strictly horizontal), steplike solifluction lobes or meltwater channels | Indicative of former lake levels of ice-dammed lakes, possible to infer the location of the ice margin | Polyline midway between toe and inner break of shoreline | Jansson (2003); Stroeven et al. (2016); Regnéll et al. (2019) |
| **Perched delta** | Flat top surface and steep delta front, situated above present lake levels. Signs of erosion by streams. **(Fig. 4b)** | Hundreds of meters wide | May be confused with an ice-contact delta, although these often have kettle holes | Indicative of former lake levels of ice-dammed lakes, possible to infer the location of the ice margin | Polygon demarcating flat top surface | Jansson (2003); Margold et al. (2013); Peterson and Smith (2013); Goodship and Alexanderson (2020) |
| **Outlet channel** | Channel at the lowest point of a water divide or along valley slopes. Often associated with washed bedrock zones. **(Figs. 5a–d)** | Tens to hundreds of meters long, tens of meters wide | May be confused with proglacial or lateral meltwater channels, although outlet channels are typically larger | Indicative of the threshold of former ice-dammed lakes | Polyline along thalweg of channel, arrow pointing downslope | Jansson (2003); Regnéll et al. (2019) |

stages were isolated and the outlines of the ice-dammed lakes were drawn by tracing the corresponding contour line of the

180 stage. Interpolation between the contour lines that corresponded to the elevation range of the stage was necessary to account for the tilting. The ice margin that constituted the obstruction damming each lake stage was placed to fit the distribution of

ice-dammed lake traces. The volumes of the ice-dammed lakes, approximated by the difference between the ice-dammed lake surface and the present day lake surface (DEM), were calculated in Python 3.9.18 following the methodologies of Regnéll et al. (2023). The difference in volume of two consecutive ice-dammed lake stages represents the approximation of the outburst flood
volume.

## 4 Results

The glacial geomorphology of the Torneträsk Basin is presented in Fig. S1. The total comes to 6633 mapped features, of which there are 2796 lineations, 678 esker segments, 39 areas of ribbed moraine, 1262 meltwater channels, 155 marginal moraines, 510 undifferentiated moraines, 894 raised shorelines, 206 perched deltas, 25 outlet channels, and 38 areas of veiki moraine.
Note that the count includes all segments of a landform, so it represents a feature count instead of a landform count.

### 4.1 Subglacial landforms: Ice flow direction

#### 4.1.1 Lineations

Lineations occur across the area, but are most common in the premontane region (60%, Fig. S1). The lineations are often in the form of a thin drumlinized till cover, and are typically hundreds of meters in length. Large-scale drumlins (thousands
of meters in length) are rare. There are two dominant ice flow directions that can be inferred from the lineations, namely towards the northeast (NE) and southeast (SE). Ice flow towards NE is the most frequent orientation, where the lineations are orientated predominantly NNE in the north, but gradually change towards a NE flow direction in the south. These lineations are exclusively in the typical size-range (Table 1). The SE ice flow direction is mostly represented by large-scale drumlins. At a few locations these two sets of lineations are cross-cutting each other, yielding that NE lineations are superimposed on the
larger SE drumlins and are younger (Fig. 3a).

#### 4.1.2 Eskers

Eskers occur across the area, but are most frequent in the montane region (63%). There, the eskers mainly trace the valley floors parallel to the valley axis. The majority of the eskers are fragmented pieces of tens to hundreds of meters in length, but they can be traced over distances of several kilometers (Fig. 3b). The fragments are mostly ridge-like, but can also be short and
205 almost circular, in which case they represent esker beads (Livingstone et al., 2020). Most eskers, often also traceable over the longest distances, parallel a NE direction. There exist a few eskers, frequently degraded, that display a SE orientation.

#### 4.1.3 Subglacial meltwater channels

Meltwater channels are the second-most abundant landform in the study area. They occur most densely in the eastern part of the montane region, while meltwater channels are almost completely lacking in the scoured bedrock zone in the west (Fig. S1).
Subglacial meltwater channels are prevalent in the entire study area, although most of them occur in the premontane region

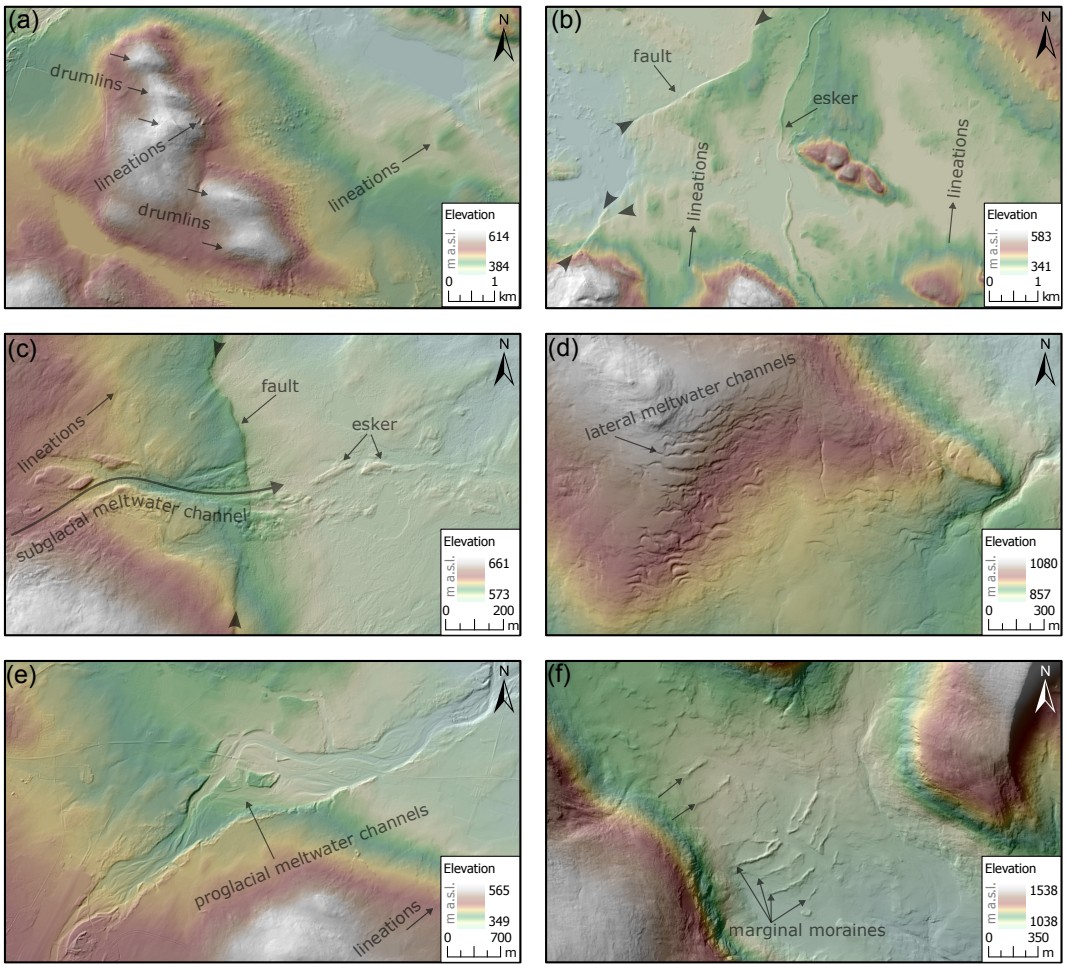

**Figure 3. Examples of subglacial, ice-marginal, and proglacial landforms in the study region.** Background is a shaded relief from the DEM provided by ©Lantmäteriet. (a) Lineations overprinting drumlins oriented at a different angle. (b) Esker aligned with lineations. The Pärvie Fault is cutting the lineations. (c) Lineations cross-cut by a subglacial meltwater channel, which transitions into esker fragments downstream. Only the deepest channel cuts through the Pärvie Fault scarp. (d) A series of lateral meltwater channels sloping towards the east. (e) Proglacial channels sloping northeast. (f) A series of cross-valley moraines. See locations of examples a–f in Fig. 1c and consult Fig. S1 for symbology.

(72%, Fig. S1). Subglacial channels are frequently situated on valley floors or valley sides (Fig. 3c). The aspect of the channels varies with local topography, but the majority of the channels indicate water flow (and therefore ice flow) towards the NE. The dimensions of the channels vary, but they are often relatively short (few hundreds of meters), considerably shorter than other types of channels (up to kilometers).

### 4.1.4 Ribbed moraine

Ribbed moraine occurs predominantly in the premontane region (56%) and in the montane region on uplands north of Torneträsk and in-between Rautasjaure and Torneträsk (Fig. S1). Ribbed moraine is located on relatively flat surfaces, either as part of valley bottoms or on uplands. They are often situated in long and narrow zones next to lake basins, which leads to a suspicion that ribbed moraine also exists on the bottom of the adjacent lakes.

Ribbed moraine ridges are generally oriented perpendicular to lineations. Indeed many ridges have a NW-SE orientation and correspond to a NE ice flow direction. In contrast, the north-westernmost occurrences of ribbed moraine show ridges with a N-S orientation, which corresponds to an ice flow direction roughly parallel to the long axis of Torneträsk, although it is unclear whether it concerns westerly or easterly flow. Sporadically, the ribbed moraine has been fluted, usually perpendicular to the ribbed moraine ridges themselves. Additionally, meltwater channels occasionally cut into ribbed moraine.

## 4.2 Ice marginal landforms

### 4.2.1 Lateral meltwater channels

Whereas subglacial channels are abundant in the premontane region, lateral meltwater channels are relatively rare (21%). Instead, lateral meltwater channels usually occur on steep valley slopes in the montane region (Fig. 3d), and relatively often on east-facing slopes. The majority of the lateral meltwater channels are formed by water flow (and ice flow) towards north or east, which indicates a general ice margin retreat towards the south or west. However, because of the steep topography in the mountains, lateral meltwater channel slope directions vary considerably.

### 4.2.2 Proglacial meltwater channels

Few proglacial meltwater channels were confidently mapped (Fig. 3e). The reason for this is probably that their identification criteria (Table 1) are insufficiently distinct to distinguish them from contemporary channels (Greenwood et al., 2007). While subglacial or lateral meltwater channels often occur upstream of the proglacial channels, outwash plains often occur in close association to them.

### 4.2.3 Moraine

There are relatively few ice-marginal moraines in the study area (Fig. S1). Most moraines are found in the vicinity of the highest mountain peaks south of Torneträsk, close to the margins of contemporary glaciers or in ice-free valleys (Fig. 3f). The orientation of the moraines in the montane region is therefore variable. Only a few marginal moraines are unambiguously associated with ice sheet configurations (Heyman and Hättestrand, 2006). These latter predominantly occur as relatively straight, sub-horizontal ridges paralleling valley sides. However, some moraine ridges have an irregular, sinuous, morphology and are then referred to as complex moraines (Heyman and Hättestrand, 2006). These complex moraines are considered to be deposited by

the ice sheet as ice-marginal moraines, but deformed through mass movements on steep slopes following deglaciation (Heyman

and Hättestrand, 2006).

Moraines are virtually lacking in the premontane region (4%, Fig. S1), except for some groups of parallel, sharp-crested, lobate ridges, mapped as "undifferentiated moraines" as their width is on average smaller than other ice-marginal moraines. Those ridges have an approximately E-W orientation, which is at an almost perfect perpendicular angle to the youngest lineations in the vicinity.

## 4.3  Ice-dammed lake traces

### 4.3.1  Raised shorelines

Raised shorelines are widespread along Torneträsk, Rautasjaure (Fig. 4), and in several valleys that hosted smaller lakes. The individual shoreline segments are usually hundreds of meters in length, but are ultimately traceable over distances of tens of kilometers. The shoreline width appears to vary with the steepness of the slope, usually meters in width on steep slopes but

tens of meters in width on gentle slopes. The shorelines appear best developed on south- and west-facing slopes of Torneträsk, which is especially clear in the middle part of the basin. The number of raised shorelines along ice-dammed lake Torneträsk (IDLT) decreases towards the southeast, with its southeasternmost occurrence along Torneälven, a mere 10 km northeast of Kiruna (Figs. 4d, 1c).

### 4.3.2  Perched deltas

Perched deltas predominantly occur on the north-facing slopes above ice-dammed lakes Torneträsk and Rautasjaure (Fig. S1). This is consistent with ice retreat towards the south and sediment-rich proglacial drainage towards the north. The perched deltas usually occur inset at successively lower elevations at the end of valleys or chutes (Fig. 4b). Perched deltas are often laterally connected with raised shorelines (Fig. 4b). Occasionally, an esker connects to the apex of a delta, strengthening its glacial meltwater origin.

### 4.3.3  Outlet channels

Outlet channels formed in association with consecutive ice-dammed lake stages (Fig. 5). The channels occur both as spillways following the natural topography below the lowest cols in water divides (Fig. 5b, c), or as lateral outlet channels on valley sides guided by the ice margin forming the barrier (Fig. 5d). Elevations at the head of outlet channels match with elevations of nearby raised shorelines and perched deltas of the corresponding ice-dammed lake stage (Fig. 6a).

### 4.3.4  Ice-dammed lake stages of Torneträsk

Numerous ice-dammed lake stages can be distinguished in the Torneträsk Basin (Fig. 6a, Table 2). Figure 6a shows that raised shorelines, perched deltas, and outlet channels plot along elevation gradients reflecting post-formation differential glacio-isostatic uplift. These gradients range from 0.5–0.7 $\mathrm{m km^{-1}}$ (Table 2). Synglacial or postglacial movement on the Pärvie Fault

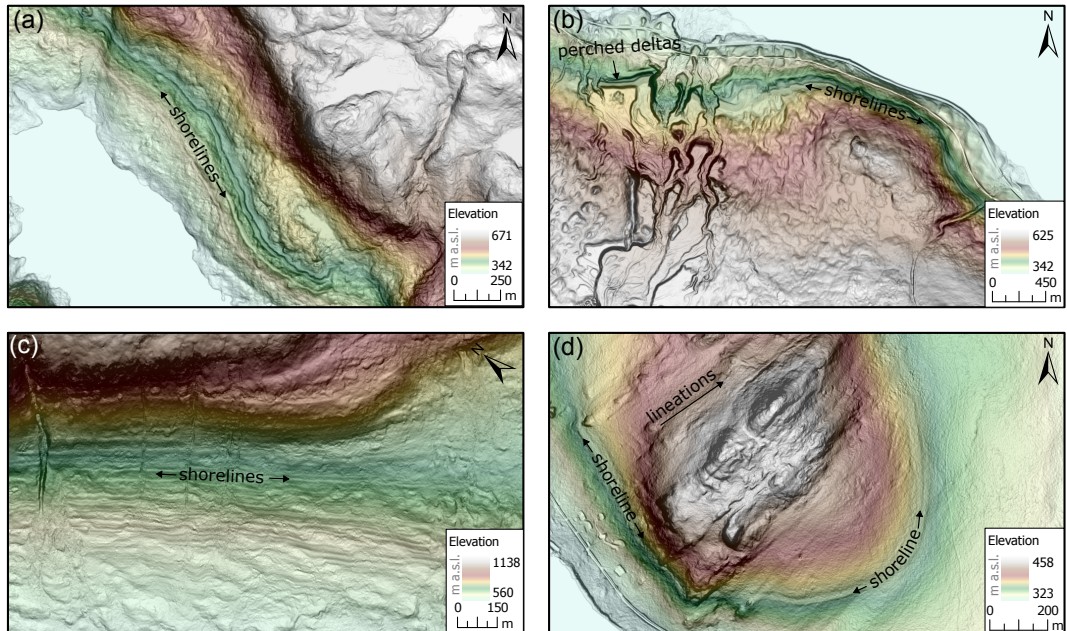

**Figure 4. Examples of landforms associated with ice-marginal lakes in the study region.** Background is a slope-relief model based on the DEM provided by ©Lantmäteriet. (a). Raised shorelines of several stages of ice-dammed lake Torneträsk. (b) Perched deltas and matching raised shorelines. (c) Raised shorelines of several stages of ice-dammed lake Rautasjaure. (d) Southernmost shoreline of the lowest lake level (youngest stage) of ice-dammed lake Torneträsk overprinting lineations at Alanen Kallovaara. See locations of examples a–d in Fig. 1c and consult Fig. S1 for symbology.

(Figs. S1, 3b and 3c) is expressed by an up to 8 m throw between shoreline segments on either side of the fault trace, thus
raising the need to compare gradients east and west of the fault trace with those that were undeformed (younger shoreline generations, Table 2, Fig. 6b).

For IDLT, eight stages are identified (T1 to T8), with the highest lake stage at least 143 m above the current lake level of 342 m a.s.l. and the lowest lake stage partially extending below the current lake level due to landscape-tilting by differential post-glacial rebound (Fig. 6). Lake stages T3 to T6 appear to be affected by the Pärvie Fault, as gradually-changing shoreline
elevations suddenly jump in elevation by 5–8 m at the location of the fault trace (Fig. 6). However, whereas there is abundant information on fault displacement of glacial geomorphology, indicating that the Pärvie Fault ruptured after landform formation (Figs. 3b and 3c), there are no geomorphological cross-cutting relationships visible in the LiDAR imagery that show the offset of raised shorelines at the exact location of the fault scarp (Fig. 6b).

Ice-dammed lakes T1 and T2 do not record the offset by the Pärvie Fault due to the lower resolution of the data (Fig. 6a).
Moreover, shorelines with relatively high elevations (i.e., above 500 m a.s.l.) that are situated above the northern shore of Torneträsk are more weakly developed and spatially limited than lower shoreline traces. Hence, they do not appear to indicate the presence of large open lakes during the generation of T1 and T2 shorelines, but rather smaller ice-marginal lakes that

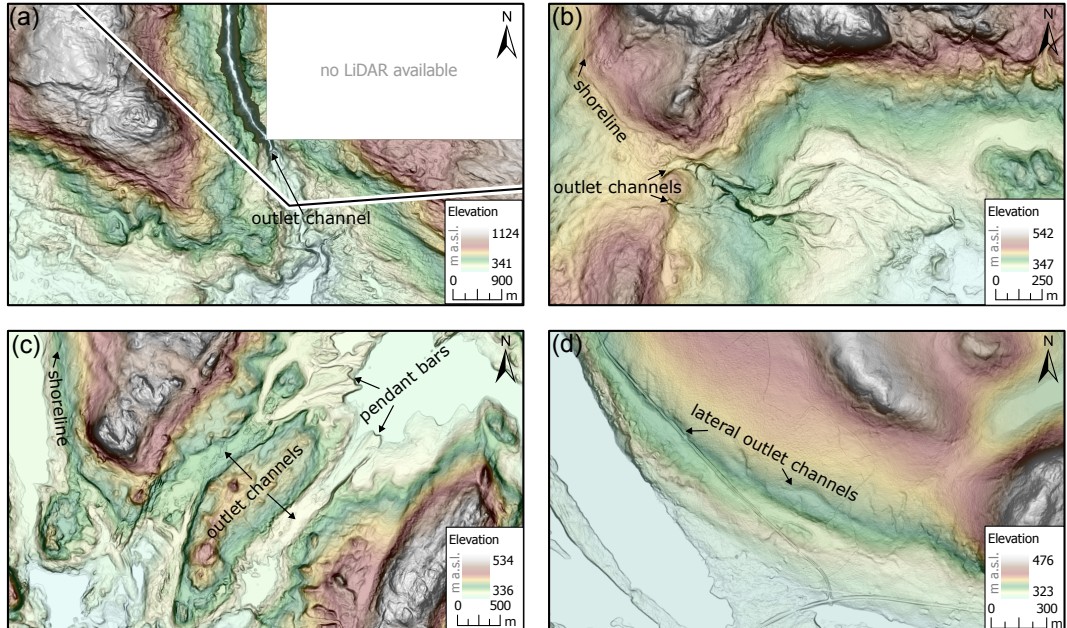

**Figure 5. Examples of outlet channels associated with ice-dammed lakes in the study region.** Background is a slope-relief model based on the DEM provided by ©Lantmäteriet. (a). A canyon (Sördalen, Norway) that served as an outlet channel for ice-dammed lake Vassijaure and spillway of the highest level of ice-dammed lake Torneträsk. (b) Incised outlet channels associated with upstream raised shorelines. (c) Two outlet channels (subsequent spillways) with pendant bars, indicating flood events. (d) Lateral outlet channels indicating final drainage of ice-dammed lake Torneträsk. See locations of examples a–d in Fig. 1c and consult Fig. S1 for symbology.

**Table 2.** Overview of the eight ice-dammed lake stages of Torneträsk identified from the elevations and distribution of raised shorelines, perched deltas, and outlet channels. The elevation ranges between highest and lowest shoreline trace are derived from the identified ice-dammed lake stages in Fig. 6. Volumes were calculated only for the ice-dammed lakes outlined in Fig. 7.

| Stage | Elevation range (m a.s.l.) | Gradient (mkm$^{-1}$) | Elevation range* (m a.s.l.) | Gradient* (mkm$^{-1}$) | Lake area (km$^2$) | Lake volume (km$^3$) | Flood volume (km$^3$) |
|---|---|---|---|---|---|---|---|
| T1 | 485–520 | 0.7 | | | | | |
| T2 | 471–495 | 0.7 | | | 530 | 55 | 1 |
| T3 | 416–442 | 0.6 | 447–462 | 0.5 | 789 | 54 | 15 |
| T4 | 407–432 | 0.5 | 440–452 | 0.5 | | | |
| T5 | 391–416 | 0.5 | 423–432 | 0.5 | 748 | 39 | 12 |
| T6 | 372–397 | 0.6 | 403–417 | 0.5 | 706 | 27 | 19 |
| T7 | 351–382 | 0.5 | | | 504 | 7 | 2 |
| T8 | 347–369 | 0.5 | | | 484 | 5 | 5 |

*Shorelines that were offset by the Pärvie Fault (T3–T6) have gradient calculations for both their western and eastern segments.

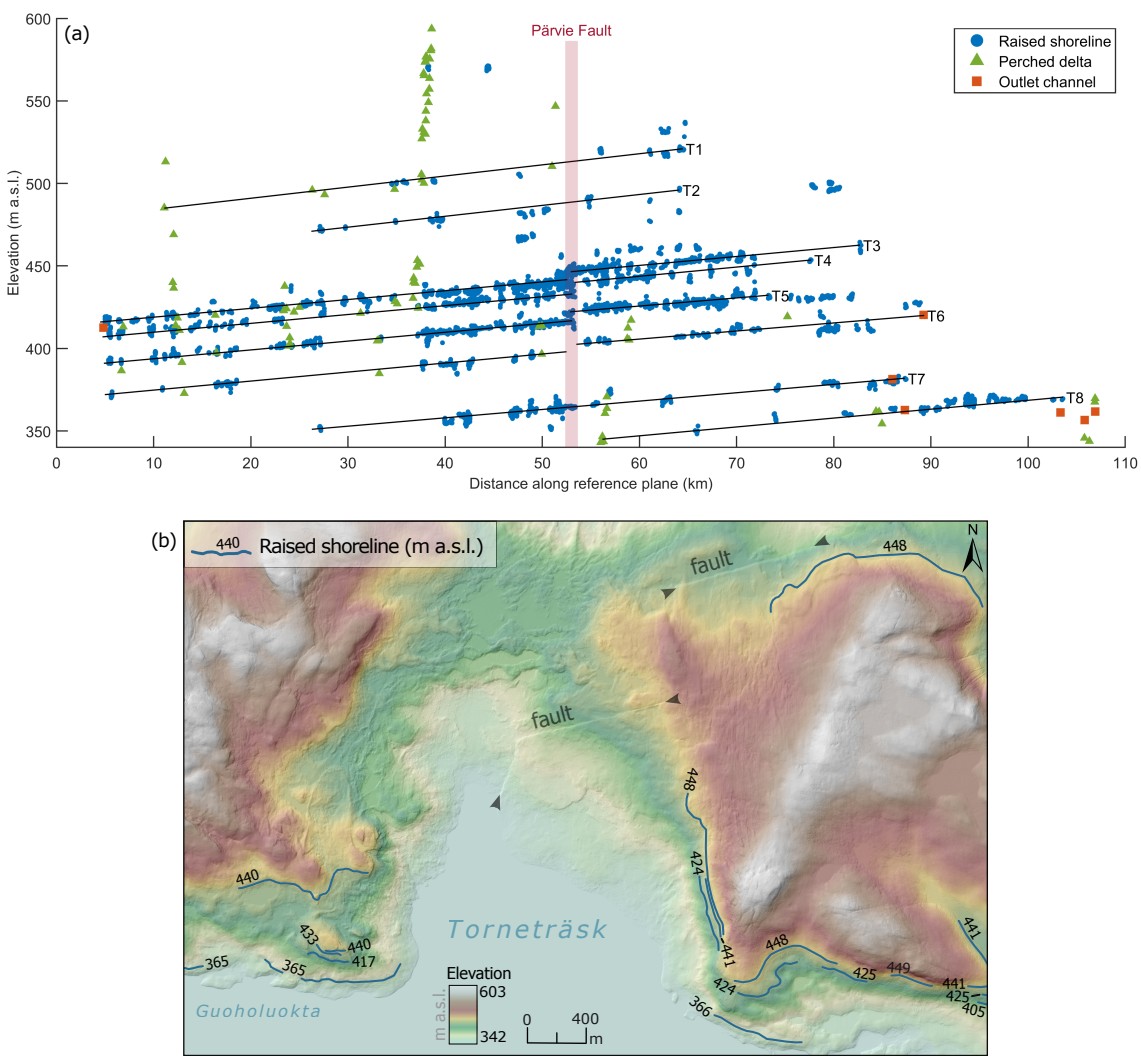

**Figure 6. Lake stages of ice-dammed lake Torneträsk.** (a) Individual lake levels were identified from the elevations of raised shorelines, perched deltas, and outlet channels. At the Abscissa value of zero, the ordinate value is 342 m a.s.l., the current elevation of the surface of Torneträsk. The approximate location where the Pärvie Fault cross-cuts the Torneträsk Basin is indicated by the red bar. The distance is calculated along an axis perpendicular to the isobases of postglacial rebound of the shorelines (see Fig. 1c). The corresponding elevation ranges are summarised in Table 2. (b) Elevations of raised shorelines of ice-dammed lake Torneträsk on either side of the Pärvie Fault where it cross-cuts the northern shore of Torneträsk (see red bar in (a) and Fig. 1c), illustrating elevation jumps of around 8 m for the higher raised shorelines (T3–T6), while the lowest raised shoreline (T7) crosses the fault at 365–366 m a.s.l. The background is a shaded relief based on the DEM provided by ©Lantmäteriet. See location in Fig. 1c.

existed between a large ice lobe in Torneträsk and the adjacent valley slope. The shorelines of ice-dammed lake T7 are clearly

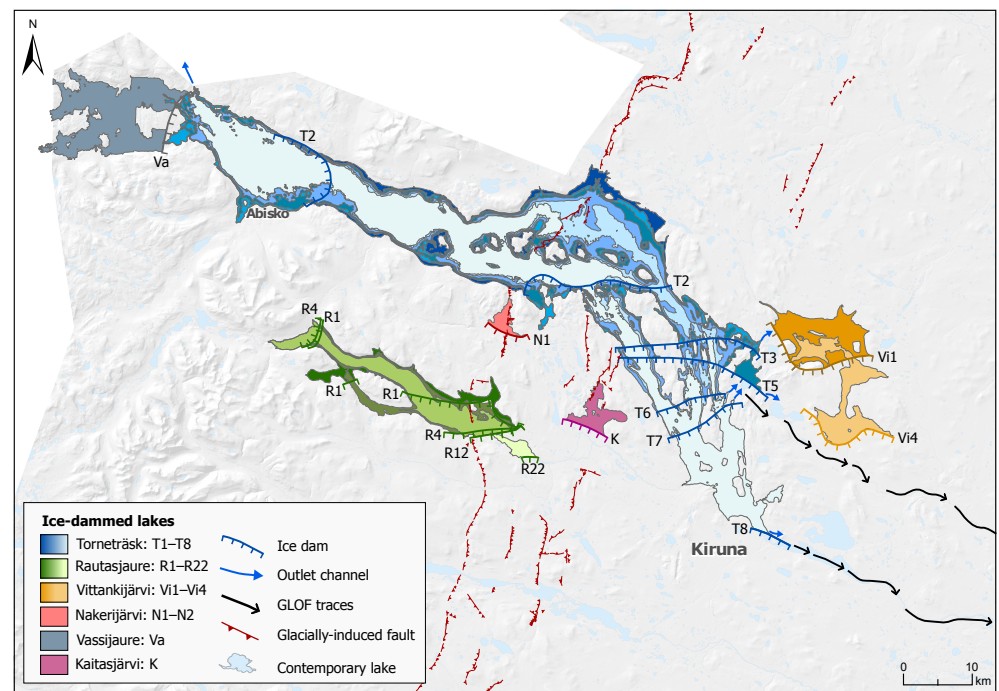

**Figure 7. Reconstruction of the ice-dammed lakes in the study area.** This figure illustrates the extents of different stages of ice-dammed lake Torneträsk and neighbouring lakes, their damming ice-margins, and outlet channels. See Fig. 6 and Table 2 for the elevations of the ice-dammed lakes of Torneträsk. The background is a hillshade based on the DEM provided by ©Lantmäteriet. Glacially-induced faults from Munier et al. (2020). Evidence for, and impacts of, glacial-lake outburst floods (GLOFs) related to IDLT stages 6-8 are presented in Fig. 8.

unaffected by the Pärvie Fault displacement, while shorelines of ice-dammed lake T8 dive below the current lake level at the location of the fault, yielding that the fault ruptured between ice-dammed lakes T6 and T7.

The ice-dammed lake traces also indicate several stages for ice-dammed lakes Rautasjaure, Vittankijärvi, Nakerijärvi, Vassijaure and Kaitasjärvi (Fig. 7). However, the distribution of shorelines belonging to these lakes is spatially limited. For ice-dammed lake Rautasjaure (Fig. 4c), 22 stages were identified, with the highest lake stage around 274 m above the current lake level of 560 m a.s.l.

### 4.3.5 Glacial lake outburst flood (GLOF)

The morphology of mapped outlet channels of IDLT indicates that only relatively small drainage events characterized the transition from one stage to the next for the oldest five lake stages. Although ice-dammed lakes T3 and T5 had flood volumes of respectively 15 and 12 $km^3$, the effect of those volumes of water disappear along the pathway. Downstream of the outlet channels of the lowest three stages (T6–T8) are traces of outburst floods (Fig. 8). For example, the lowest ice-dammed lake stage of Torneträsk (T8) is associated with initial drainage through two lateral channels paralleling the current river Torneälven (Fig. 5d), but ultimately results in a drainage event in the form of an outburst flood of 5 $km^3$ (Fig. 8, Table 2).

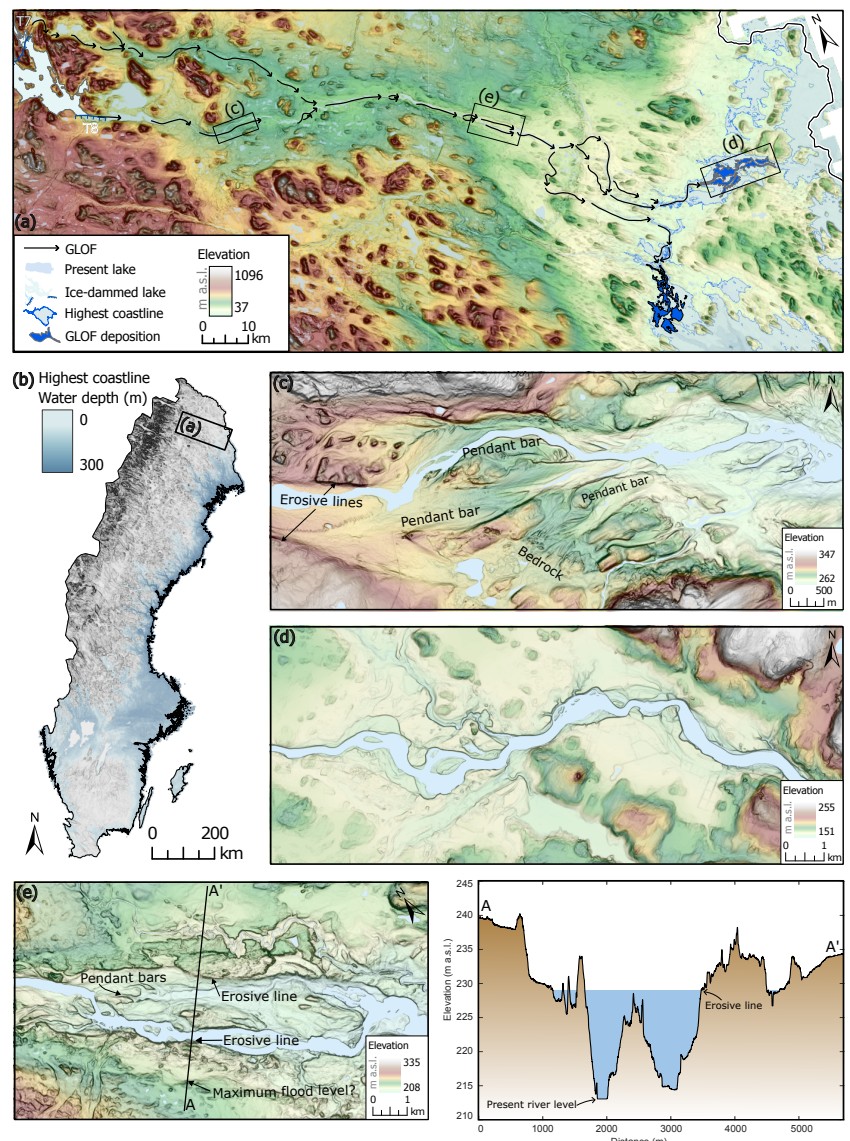

**Figure 8. Traces of glacial lake outburst floods (GLOFs).** Background to panels a–e is a slope-relief model based on the DEM provided by ©Lantmäteriet. (a) Pathways of the GLOFs that drained ice-dammed lake Torneträsk (T6–T8) in stages through Tornedalen and deposited sediments in the Baltic, which, around the time of its highest postglacial incursion (as indicated by the highest coastline; light blue polygons), was a lake; Ancylus Lake. Highest coastline and GLOF deposits taken from datasets provided by the Geological Survey of Sweden (SGU, 2024a, b). (b) Extent of postglacial incursion as delineated by the highest coastline. This is a time-transgressive imprint, the age of which becomes younger towards the north. Boxes outline the research area and the extent of panel a. (c) Pendant bars in the lee of (bedrock) obstacles. (d) Delta deposits where the GLOFs entered Ancylus Lake. (e) Erosive lines indicating GLOF flooding levels, as visible in cross section A–A'.

Unambiguous geomorphological traces of an outburst flood (cf. Wells et al., 2022), presumably initiated during the largest of drainage events (T6, 19 km$^3$; Table 2), in the form of pendant bars (Fig. 8c), erosional trimlines (Fig. 8e), channel incision, and scoured bedrock, are on display along Torneälven for 115 km. The flood path also submerged a shallow interfluve, c. 95 km downstream from the initial drainage location(s), establishing a unique river bifurcation that separates drainage through Tornedalen from Kalixdalen by means of the river Tärendöälven. Mapping of surficial geology by the Geological Survey of Sweden (SGU, 2024a), carried out from the 1980s to early 2000s based on aerial imagery supported by sparse field observations, shows that vast amounts of glacio-fluvial sediments which form deltaic deposits are found 165 km downstream of the initial drainage location(s) along both Tornedalen and Kalixdalen (Fig. 8d). Today, Torneälven proceeds towards the east, where it curves towards the south, straddling the border with Finland, until it reaches the Baltic Sea at Haparanda. Tärendöälven immediately flows southwards into Kalixälven, and also ultimately exits into the Baltic Sea (at Kalix).

## 5 Discussion

### 5.1 Ice-dammed lake evolution

Several ice-dammed lake systems were in operation during deglaciation in basins currently occupied by Torneträsk, Rautasjaure, Vittankijärvi, Nakerijärvi, Vassijaure, and Kaitasjärvi (Fig. 7). The identified ice dams are consistent with the distribution of shorelines, perched deltas, and outlet channels, are required to impound the lakes against surrounding topography, and are consistent with the orientation of meltwater channels, lineations, and eskers. Those stages of IDLT for which a plausible outlet could be identified are outlined in Fig. 7 and described in the following sections.

Because lake Vassijaure (461 m a.s.l.) is located at the far western end of the Torneträsk Basin, ice-dammed lake Vassijaure (Va; Fig. 7), with perched delta elevations between 510 and 513 m a.s.l., is included in this subsection. Its spillway, located at 510 m a.s.l., directed meltwater through the valley Norddalen towards Rombaksfjorden in Norway. A set of perched deltas at the head of the fjord are likely related to the incision of the spillway, and indicates 95 m of relative sea level lowering due to glacio-isostatic uplift. Ice-dammed lake Vassijaure required an ice dam just west of Sördalen (Fig. 7), and was in operation until ice retreat exposed Sördalen, a much lower passage at 412 m a.s.l. into Norway (Fig. 5a). Multiple channel incisions and a general lack of sediment along the western Sördalen valley side demonstrate the erosive power of the drainage of ice-dammed lake Vassijaure, as the lake level dropped about 50 m.

IDLT stage T2, with raised shoreline elevations between 471 and 495 m a.s.l., required two specific ice margin locations. One ice margin prevented the lake to drain westwards through Sördalen (Figs. 5a and 7). Given the requirement for an ice dam in western Torneträsk Basin abutting the northern shore of Torneträsk and preventing the drainage of both ice-dammed lakes Vassijaure and Torneträsk (T2) through Sördalen (Fig. 7), the most plausible configuration yields a piedmont lobe emanating from Abiskodalen. The other ice sheet margin obstructed drainage across lower terrain towards the southeast. This and subsequent damming margins (T3 to T8) outline the retreat of a large coherent ice body towards the south-southwest. A spillway towards Vuoskujärvi, a lake north of the study area (not shown), was controlling the highest elevation of the lake to 500 m

a.s.l. Ultimately, T2 drained through Sördalen as the ice lobe retreated from the northern shore, and the Sördalen outlet channel
becomes the spillway of the next lake stage (T3).

Ice-dammed lake stage T3, with raised shoreline elevations between 416 and 462 m a.s.l., presumably had its spillway through Sördalen at 412 m a.s.l. (Fig. 5a). Because the lowest shorelines in the vicinity, with elevations of 417 m a.s.l, are several meters above the current elevation of the spillway, this potentially reflects deepening of the spillway by c. 5 m through continuous runoff. A coherent ice margin in the southeast had to plug two valleys, requiring an ice dam with an E-W orientation
(Fig. 7). An E-W ice margin is also justified from the presence of nearby eskers and lineations with orientations perpendicular to the ice margin. Morphological traces of a drainage event (15 km$^3$, Table 2), that lowered the lake level to the next stage, indicates drainage routing to nearby ice-dammed lake Vittankijärvi (Vi, Fig. 7). To accommodate this drainage route the ice margin of IDLT stage T3 was most likely connected to the ice margin of one of the lower stages of ice-dammed lake Vittankijärvi to expose terrain between both lakes (i.e., between Vi1 and Vi4, Fig. 7).

The ice front impounding IDLT stage T5, with raised shoreline elevations between 391 and 432 m a.s.l., probably had a rather similar configuration as the ice margin reconstructed for ice-dammed lake stage T3 (Fig. 7). The ice sheet retreated around 5 km, although slightly more in the east than in the west. IDLT stage T5 had its outlet channel at 424 m a.s.l. with fluvial deposits downstream of the channel over a distance of 1.5 km (Fig. 5b) and drained 12 km$^3$ of water (Table 2) to ice-dammed lake Vittankijärvi (Fig. 7). Drainage routing into the lake explains the absence of traceable GLOF imprints.

IDLT stage T6, with raised shoreline elevations between 372 and 417 m a.s.l., drained through an outlet channel with a width of approximately 750 m at 383 m a.s.l. (see western channel in Fig. 5c). GLOF traces occur downstream of the outlet channel in the form of streamlined pendant bars on the lee side of bedrock protrusions (Fig. 5c; northern pathway in Fig. 8a). Indeed, when the ice dam failed, 19 km$^3$ of lake water (Table 2) was routed through Tornedalen.

IDLT stage T7, with raised shoreline elevations between 351 and 382 m a.s.l., required a similar ice margin configuration
as for IDLT stage T6 with their outlet channels a mere 1 km apart (Fig. 7). The lake level coincided with the elevation of the outlet channel of the previous stage at 383 m a.s.l., now acting as a spillway (western channel in Fig. 5c). The outlet channel at 365 m a.s.l. (eastern outlet channel in Fig. 5c) has an internal topography of c. 5 m. Erosion of at least 5 m appears to have occurred due to the drainage of IDLT stage T7 (2 km$^3$, Table 2), thus establishing a new spillway altitude for IDLT stage T8 at 365 m a.s.l., and providing a source of material for downstream GLOF deposits (blue polygons in Fig. 8a, flat terrain in Fig.
8d).

The lowest lake level of IDLT, stage T8, with shorelines situated between 347 and 369 m a.s.l., was dammed approximately 18 km south of T7 (Fig. 7). The lake level of T8 was probably controlled by the outlet channel of T7 (Figs. 7, 5d). Drainage of ice-dammed lake T8 occurred predominantly through Tornedalen. Two drainage channels paralleling the current river Torneäl- ven occur at 365 m a.s.l. and 357 m a.s.l. (Fig. 5d), which appears to demonstrate that the drainage of 5 km$^3$ occurred in
three steps: initial drainage along the ice margin, another drainage after thinning of the ice sheet, and a final drainage as all of Tornedalen opened up (Table 2).

Extrapolation of the available data on the elevations of ice-dammed lake traces to determine the western extent of IDLT stage T8 is hampered by invisibility (Fig. 6a). Indeed, the absence of shoreline traces west of the Pärvie Fault is most likely due to

their submergence as a result of differential landscape uplift (creating the illusion of lake tilting relative to the raised shorelines).

Ice-dammed lake T8 shorelines would be predicted at elevations at or above approximately 325 m a.s.l. in this sector, which is up to 17 m below the current lake level of Torneträsk. Bathymetric data is therefore required to faithfully outline the perimeter of T8, but the most recent open-source bathymetric map is based on measurements from 1920–1921 (SMHI, 2020). Nevertheless, the available bathymetric information yields that the 20 m bathymetric contour occurs at a relatively short distance inboard from the current shoreline, indicating a rather steep basin. Hence, the current shoreline of Torneträsk is therefore used as perimeter for the western part of IDLT stage T8, and therefore slightly overestimates its areal extent. Moreover, the current lake surface was used as base surface for the calculations of GLOF volumes, leading to an overestimation of the GLOF volume of stage T8 (Table 2).

Drainage of IDLT stage T6 resulted in the largest of the GLOFs with an estimated volume of 19 km$^3$ (Table 2). In comparison, a catastrophic GLOF in central Jämtland, instigated by the failure of an ice saddle between the southern and northern domes of the FIS, released an estimated 200 km$^3$ (Regnéll et al., 2023). It appears that the Torneträsk GLOFs terminated in Ancylus Lake, a freshwater lake formed by isolation following isostatic rebound, occupying the current Baltic Sea Basin (Lundqvist, 1986; Björck, 1995; Lindén et al., 2006). Raised shorelines of the highest coastline of Ancylus Lake (Figs. 8a and 8b) appear approximately 165 km downstream of the initial drainage location(s) of the GLOFs. Mapping of surficial geology by the Geological Survey of Sweden (SGU, 2024a), carried out from the 1980s to early 2000s based on aerial imagery supported by sparse field observations, shows that vast amounts of glacio-fluvial sediments which form deltaic deposits are found 165 km downstream of the initial drainage location(s) along both Tornedalen and Kalixdalen (Fig. 8d). The most parsimonious explanation for these glacio-fluvial deposits is that they originated from the debris-laden GLOFs entering Ancylus Lake at the highest coastline at an elevation of 170 m a.s.l., where discharge velocities decreased, sediments were deposited, and a deltaic landform was created.

## 5.2 Ice-marginal positions

Shrinkage of FIS across the study area (cf. Fig. 2c) yielded a complicated pattern with southeastward retreat in the far western corner (isolating ice-dammed lake Vassijaure), initial southeast-southward retreat of the FIS across the Torneträsk Basin, and eventually curving towards southwestern retreat in the southern part of the study area (Fig. 9). This overall pattern aligns with former reconstructions (Kleman et al., 1997; Hättestrand, 1998; Stroeven et al., 2002, 2016; Hughes et al., 2016) and is reflected by the orientation of the small-scale lineations, eskers, and subglacial meltwater channels in the premontane region (Fig. 1c; Fig. S1). Disentangling glacial geomorphological traces in the mountains, determining their succession, construction of swarms, and assigning absolute ages (even determining if landforms are deglacial or not), is challenging. For example, it can remain unclear whether moraines were formed by advancing valley glaciers situated up-valley of the moraine or by outlet glaciers from a thinning ice sheet (Heyman and Hättestrand, 2006). This problem is exacerbated by the knowledge that landforms in these mountains are not exclusively from the last deglaciation, including moraines (Kleman, 1992; Fredin and Hättestrand, 2002; Fabel et al., 2006; Heyman and Hättestrand, 2006). Nonetheless, the retreat in the premontane region is

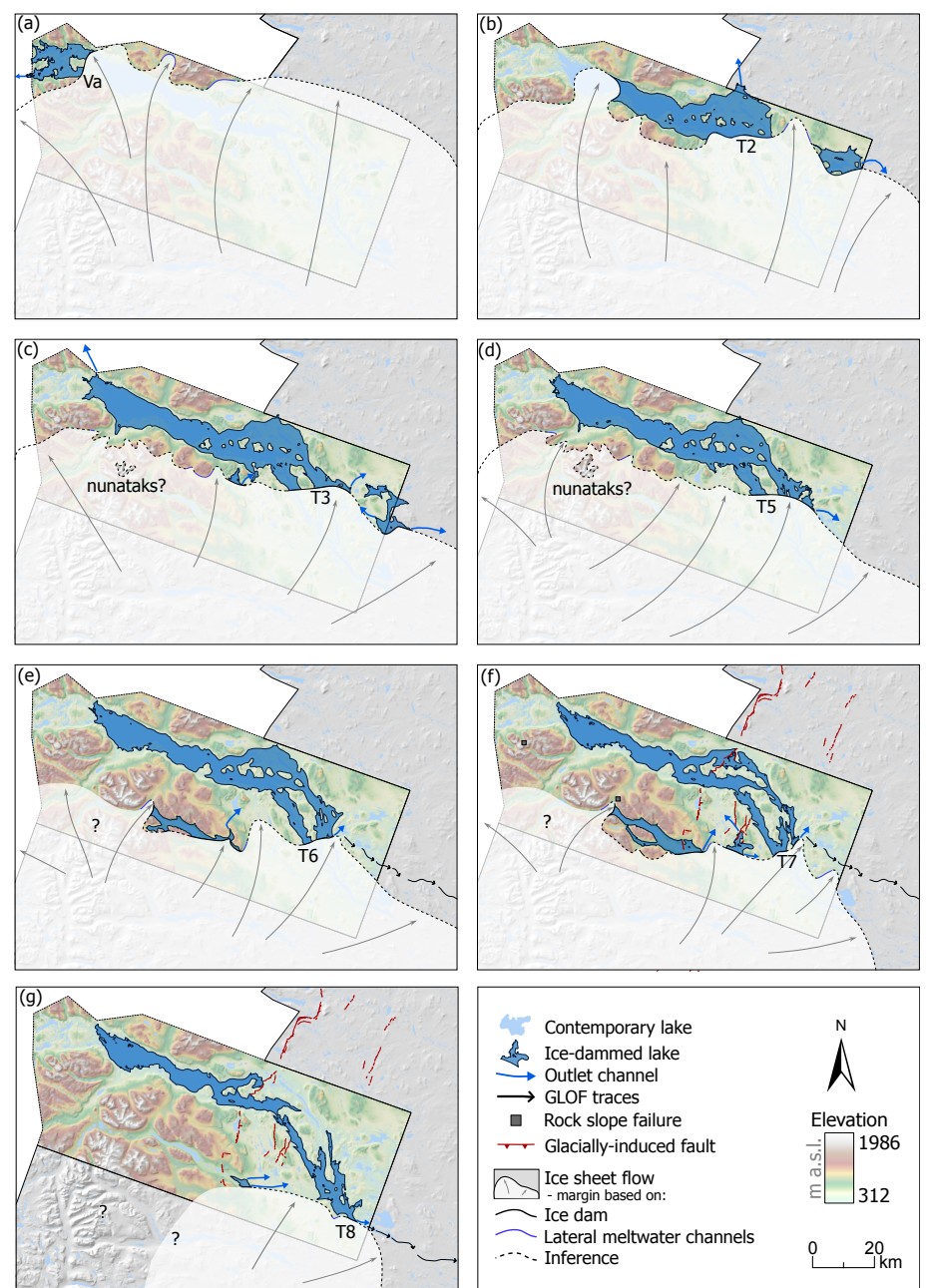

**Figure 9. Reconstruction of the ice-marginal positions during deglaciation.** Reconstruction is based predominantly on the ice dams and outlet channels of ice-dammed lakes Vassijaure (a) and Torneträsk and Rautasjaure (b–g). Except for the need to block the upstream reach of ice-dammed Lake Rautasjaure (e–f), the extent of ice across the mountains remains unconstrained (e–g). The background is a shaded relief based on the DEM provided by ©Lantmäteriet. Glacially-induced faults from Munier et al. (2020).

clear, and moraines, lateral meltwater channels, and relict shorelines in the montane region require the former existence of ice lobes (Fig. 1c; Fig. S1).

There is a clear distinction between retreat in the premontane and montane regions in terms of the configuration of the ice
sheet. In the premontane region, ice sheet retreat is orderly and the margin maintains its shape and outlines a coherent ice body (Figs. 1c; 9) as ice flow directions transition from northwest through north to northeast while the ice margin pivots around the higher topography. In the montane region, meanwhile, the ice sheet disintegrates into several ice lobes (Fig. 9). Hence, a strong control of topography on ice retreat patterns and rates is evident, as other studies have demonstrated for the FIS (Stroeven et al., 2016; Boyes et al., 2023; Szuman et al., 2024), the British-Irish Ice Sheet (Greenwood et al., 2007; Hughes et al., 2014),
and the Cordilleran Ice Sheet (Kleman et al., 2010; Dulfer et al., 2022). Bed topography becomes increasingly dominant as the ice thins (Hughes et al., 2014), hence topographic control is especially significant during ice expansion and final deglaciation. Because we cannot determine the precise disintegration of the ice sheet in the mountains, save for a few margin locations such as the up-valley ice dam required for ice-dammed lake Rautasjaure (Fig. 9e,f), we have simply characterized this area with uplands becoming ice free before adjacent valleys (e.g., Fabel et al., 2002) with "nunataks" (Fig. 9c,d) and "questionmarks"
(Fig. 9e–g). Dynamically, montane ice separated from premontane ice, leading to the damming of lakes in all major easterly valleys, including farther south (Regnéll et al., 2019).

Ice-dammed lakes influence ice sheet mass balance and ice dynamics while they amplify glacier mass loss and velocity (Krinner et al., 2004; Carrivick and Tweed, 2013; Sutherland et al., 2020; Quiquet et al., 2021; Scherrenberg et al., 2023). On the ice sheet scale, it is well-known that the FIS retreated faster in the (south)east than west due to a calving margin in the
Baltic Basin (Boulton et al., 1985; Kleman, 1992; Stroeven et al., 2016). Higher ice losses of the FIS due to the ice-dammed lakes in the Torneträsk region are consistent with the amplifying effect that ice-marginal lakes have on retreat rates (e.g., Stokes and Clark, 2004; Utting and Atkinson, 2019). Thus, the presence of the ice-dammed lakes led in part to the pivoting motion of ice retreat in this region.

## 5.3 Glacially-induced faulting

Glacial isostatic adjustment is a response of the Earth to a redistribution in ice load (Wu and Peltier, 1982). A combination of ambient tectonic and glacial isostatic adjustment-induced stresses due to ice sheet thinning and retreat during deglaciation, can cause faulting, a reactivation of pre-existing faults, and earthquakes (Moon et al., 2020; Steffen et al., 2021). The reactivated faults are referred to as glacially-induced faults. The 155 km-long Pärvie Fault is the longest glacially-induced fault in Sweden (Lagerbäck and Sundh, 2008). The fault is composed of more than 200, generally 5–10 m-high, predominantly west-facing,
fault scarps (Lundqvist and Lagerbäck, 1976; Lagerbäck and Sundh, 2008; Mikko et al., 2015).

The NNE trending Pärvie Fault cross-cuts Torneträsk and we have shown that it offsets IDLT shoreline stages T3–T6 (and presumably T1 and T2), but not T7 and (by inference) T8. Because we can now show, for the first time, that the FIS margin was positioned in-between the ice-dammed lake stages T6 and T7 reconstructed ice margins when the Pärvie Fault ruptured (Figs. 7, 9e, f), it will now be possible to combine Pärvie Fault analyses from LiDAR (e.g., Smith et al., 2021) with glacial

configurations. At the very least, the implication is that the fault ruptured within 40 km from the ice margin. In fact, if the fault ruptured along its entire length at this time, then as much as 95 km of the fault trace were ice covered at that point.

Along the length of the Pärvie Fault within the study area, cross-cutting relationships between the glacial geomorphology and fault scarp traces can be studied relative to an inferred 9.9–10 cal ka BP reactivation age of the Pärvie Fault. If all of the Pärvie Fault ruptured at once, such as is typically considered when calculating the amount of energy released, cross-cutting should post-date deglaciation north of the inferred ice sheet margin at the time of rupture (between T6 and T7, Figs. 9e, f) and pre-date deglaciation south of this ice sheet margin, as originally suggested by Lundqvist and Lagerbäck (1976). However, such a systematic relationship does not exist (Fig. 10a). In fact, our documentation yields examples of pre-deglaciation faulting north of the ice margin (Fig. 10b), postglacial faulting south of the ice margin (Figs. 3c, 10c), a few cross-cutting relationships indicating the fault ruptured multiple times (Figs. 10d, e), and sites where the relationship remains unclear (Fig. 10a).

A complicating factor is that the region in northern Sweden is known for palimpsest landscapes where traces of older glaciation may be preserved despite subsequent ice sheet overriding (Kleman, 1992; Fabel et al., 2002; Hättestrand and Stroeven, 2002). Indeed, Smith et al. (2021) illustrate that some glacial landforms cross-cut by the Pärvie Fault (farther south) are not necessarily formed during the last deglaciation, and, if so, that the expression of the fault scarp itself could have been preserved beneath ice sheets. However, glacial landforms cut by the Pärvie Fault within the study area align with reconstructed deglaciation directions and so faulting here was demonstrably postglacial.

A recurring hypothesis is that the Pärvie Fault was created (or reactivated) through a single seismic event (Arvidsson, 1996; Lagerbäck and Sundh, 2008; Vogel et al., 2013). If true, the formation of the Pärvie Fault would require a thrust earthquake with a moment magnitude of $M_w \approx 8.0$ (e.g., Lindblom et al., 2015). Currently, such an approach appears unrealistic given the mounting evidence for different types of cross-cutting relationships (Fig. 10), reinforcing alternative interpretations that the Pärvie Fault ruptured multiple times. Supporting this interpretation, Smith et al. (2018) found that a fluvial terrace truncated by the Merasjärvi Fault scarp, another glacially-induced fault southeast of the study area (67.520833° N, 21.941944° E), displayed multiple ruptures.

Hoppe and Melander (1979) describe a situation which resembles that of Torneträsk where a section of the Pärvie Fault north of Lake Langas cross-cuts an esker (67.430859°N, 18.711962°E) and a series of poorly-preserved raised shorelines (67.424151°N, 18.701256°E), except for the lowest shoreline which is not displaced by the fault (Hoppe and Melander, 1979; Lagerbäck and Witschard, 1983). This setting also appears to satisfy rupture immediately after local ice margin retreat from the area (Regnéll et al., 2019). Therefore, this segment of the Pärvie Fault must have ruptured later than the rupture in the Torneträsk Basin between IDLT stages T6 and T7 because Lake Langas would have been ice covered at that time.

Multiple ruptures of the Pärvie Fault appear to explain best the observed cross-cutting relationships (Fig. 10). The shorelines of IDLT stages T3–T6, which are cut by the Pärvie Fault, are followed by IDLT stages T7 and T8, which are not offset, hence providing a geomorphological marker for the timing of this rupture of the Pärvie Fault (Figs. 7 and 9). Locations demonstrably underneath the ice sheet during the IDLT stages 6–7 Pärvie Fault rupture also exhibit cross-cutting relationships indicating at least some fault scarp segments ruptured subaerially (Figs. 9e, f, 10c). The most inclusive explanation is therefore that the fault ruptured several times within the study area: once shortly after the formation of IDLT stage T6 shorelines and north of the ice

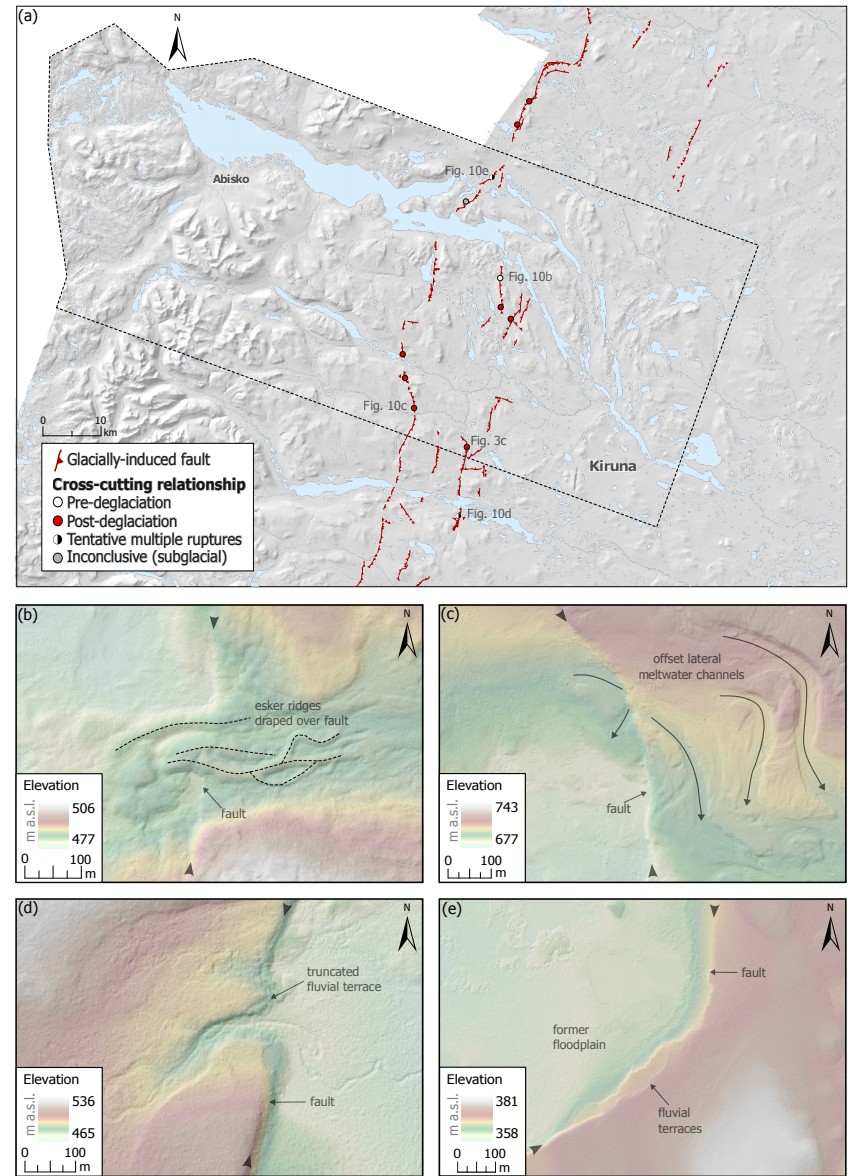

**Figure 10. Inferred cross-cutting relationships between geomorphology and the Pärvie Fault.** (a) Compilation of sites within the study area of complex relationships between expressions of the Pärvie Fault and landforms including faulting (b) pre-dating deglaciation, where an esker drapes a fault scarp, (c) post-dating deglaciation, where glaciofluvial landforms are cut by a fault scarp, and (d-e) occurring, tentatively, multiple times, where fluvial terraces are offset by multiple ruptures. Panel (d) portrays the same location as in Smith et al., 2021, Fig 12.4. The background is a shaded relief based on the DEM provided by ©Lantmäteriet.

margin at T7, and once or several times subsequently as evidenced from postglacial cross-cutting relationships south of IDLT stages T7 and T8 (Figs. 9f, g, 10). Clearly, this latter faulting south of Torneträsk did not reactivate fault traces further north,

allowing the shorelines of IDLT stage T7 to escape offsetting. The variety of cross-cutting relationships clearly illustrate the complexity and spatial reach of the re-activation of faults in response to glacial isostatic adjustment.

## 5.4 Chronology

475 Ice retreat in the Torneträsk region coincided with the formation and drainage of ice-dammed lakes and (multiple) ruptures of the Pärvie Fault. Geomorphological relationships provide insights into the relative timing of these events, but a lack of absolute ages limits our ability to precisely constrain their temporal evolution. Here, we attempt to bridge that gap by integrating existing knowledge regarding the timing of ice sheet retreat with our reconstruction of the ice-dammed lakes and faulting.

Ice retreated from the coastal zone in Norrbotten between 10.2–9.9 cal ka BP (Stroeven et al., 2016) at the same time 480 as Ancylus Lake transgressed the coastal zone, peaking at 10 cal ka BP (Lindén et al., 2006). There are no cross-cutting relationships between the GLOF-landforms and the Ancylus Lake shorelines along the floodpath. However, as pointed out, the deposition zone of GLOF sediments corresponds with the location of the locally highest coastline. The age of the GLOFs must therefore overlap with the age of Ancylus Lake (10.5–9.5 cal ka BP; Lindén et al., 2006), and, through its association with the highest coastline, can therefore perhaps be narrowed down to having occurred close to 10 cal ka BP. According to the 485 reconstruction by Stroeven et al. (2016), Tornedalen opened up approximately 9.9 cal ka BP, which is consistent with this age estimation for the GLOFs.

The shoreline gradients of Torneträsk have a tilting direction towards the northwest (325°) and decrease from 0.7 $\mathrm{mkm^{-1}}$ for the oldest ice-dammed lake stages to 0.5 $\mathrm{mkm^{-1}}$ for the youngest ice-dammed lake stages. The gradients clearly reflect the glacio-isostatic uplift pattern following the deglaciation of Fennoscandia (Steffen and Wu, 2011; Berglund, 2012). The 490 estimated gradients of Torneträsk are (i) slightly higher than the gradients of 0.4–0.5 $\mathrm{mkm^{-1}}$ that Regnéll et al. (2019) calculated for ice-dammed lakes Akkajaure and Sitojaure, located approximately 120 km farther south between the Kebnekaise and Sarek mountains and (ii) bracketed by the gradients of 0.3–0.9 $\mathrm{mkm^{-1}}$ of the shorelines of ice-dammed lakes in central Jämtland (Regnéll et al., 2023). These are favorable comparisons because ice-dammed lakes Akkajaure and Sitojaure also formed in response to the final deglaciation of the Fennoscandian Ice Sheet. Their timing is closest to the youngest GLOF 495 of Torneträsk, and so are their shoreline gradients. Additionally, lake evolution in central Jämtland spans 10.5–9.2 cal ka BP (Regnéll et al., 2023), which brackets ice retreat from the Torneträsk Basin, rendering it reasonable that the gradients of IDLT fall within the range of values from central Jämtland. We are, however, hesitant to put too much confidence in the derived gradients and comparisons between the regions. The calculations depend on the direction of the tilt along which they were calculated, the precision of the mapping of shorelines, and on post-depositional faulting.

500 In the reconstruction of Stroeven et al. (2016), the retreating ice margin swept across the study area in a time span of 500 yrs (Fig. 1b). The ice-marginal positions that dammed the successive ice-dammed lake stages of Torneträsk fall approximately in-between their 10.1 and >9.9 cal ka BP isochrons (Fig. 1b), which would suggest the ice-dammed lake system of Torneträsk existed for a total duration of <200 yrs. Shorelines are estimated to require at least a few decades to develop (Melander, 1977c; Thompson and Baedke, 1997). Waves play a key role in the development of shorelines (Lorang et al., 1993; Schuster and 505 Nutz, 2018), although in periglacial environments lake ice potentially influences shoreline development as well (Matthews

et al., 1986). Regardless, the formation of shorelines is largely dependent on the duration of exposure to shoreline-building processes, which requires a stable lake level for a certain period. Drainage of an ice-dammed lake (67.100232° N, 16.403404° E) in front of Sállajiegna, a glacier on the Swedish-Norwegian border (Jenkin, 2018), in 2013, revealed that part of a prominent shoreline was built since 2009 (within 5 years) judging from aerial imagery. This is observational evidence that a shoreline can form in mere years, not necessarily in decades. A total duration of the ice-dammed lake system of Torneträsk of 200 yrs is therefore certainly plausible. Even the shorelines of the 22 stages of the ice-dammed lake system of Rautasjaure would have been able to develop within the overall time window.

The brief period during which the ice-dammed lakes formed and drained makes it challenging to determine the role of faulting. The largest drainage, following IDLT stage T6, coincides with the fault rupture offsetting shorelines of IDLT stages T6 and older. These two events, fault rupture and drainage, appear to have happened simultaneously (i.e. 9.9–10 cal ka BP). For example, the rupture might have triggered an instability of the ice dam, causing the ice dam to fail and the lake surface elevation to lower to a new spillway level (or the other way around; the record release of water from the lake might have triggered the fault reactivation). Although there is no indisputable evidence for correlation of these two events, the implied influence of glacially-induced faulting on the stability of ice dams, and hence on the ice retreat dynamics, should not be discarded.

Dating of the Pärvie Fault could provide a minimum age for IDLT stage T6 and a maximum age for stage T7. Unfortunately, no absolute ages are available for the Pärvie Fault. However, our mapping might provide future avenues to dating the timing of fault rupture by means of dating related deposits. The drainage of IDLT stage T6 is the largest in terms of volume (Table 2) and the first to have resulted in the deposition of recognizable GLOF deposits (Fig. 8a,d). Macrofossils incorporated in basal sediments of the Torneträsk GLOF deposits at the Ancylus Lake highest shoreline could potentially yield an age for the drainage of T6. Alternatively, radiocarbon dates on macrofossils straddling the boundary between glacio-lacustrine sediments from the ice-dammed lake phase and non-glacial sediments (after drainage), in small lakes located between IDLT stages T6 and T7, could offer an additional approach to dating the T6 GLOF and, by association and within dating uncertainty, activity on the Pärvie Fault (e.g., Regnéll et al., 2023). Finally, because the outlet of IDLT stage T6 (western channel in Fig. 5c) also was the spillway channel of IDLT stage T7, bedrock in this channel would be first fully exposed to cosmic rays after failure of the IDLT stage T7 ice dam only a decade or decades after the rupture of the Pärvie Fault. Cosmogenic in situ [14]C dating of outlet channel bedrock is likely the most direct methodology to determine the age of faulting (within uncertainty). The promise of this technique to deliver accurate deglaciation ages in Sweden was demonstrated by Goodfellow et al. (2024).

Similarly, secondary deposits related to the fault rupture itself could be dated. The formation of landslides in northern Fennoscandia has been associated to earthquakes caused by post-glacial faulting (Sutinen, 2005; Lagerbäck and Sundh, 2008). There are two large RSF deposits in the study area (RSF-K and RSF-B, Fig. 1c). If they were triggered by ruptures along the Pärvie Fault, their ages could be used to date fault activity. A boulder from a RSF deposit in Kärkevagge (RSF-K, Fig. 1c; Rapp, 1960; Jarman, 2002) has been dated to $9.5 \pm 1.1$ ka BP using cosmogenic nuclide [10]Be (Stroeven et al., 2002), and a RSF scar bedrock sample (S-02-05) and boulder tongue sample (S-02-06) in Bessešvággi (RSF-B, Fig. 1c) returned [10]Be apparent exposure ages of $8.9 \pm 0.8$ ka BP and $10.9 \pm 0.8$ ka BP, respectively. All three samples are expressed using the LSDn scaling method with external uncertainties (cf. Supplementary dataset from Stroeven et al., 2016). Clearly, these results alone

are inconclusive in dating fault activity. For example, although one may argue that deep-seated (>3 m depth) bedrock in a scar should provide the most reliable age, considering that a boulder is more likely to have inherited previous exposure, we do not know from a mere two samples whether its younger age is indeed due to boulder inheritance or whether slope erosion after the main failure could have resulted in a younger apparent age of the bedrock scar. On aggregate, however, the timing of RSF activity from all three samples, within uncertainty, overlaps with our inferred timing of the Pärvie Fault of c. 9.9–10 cal ka BP.

The absence of a larger group of landslides in the vicinity of the Pärvie Fault challenges the potential earthquake-induced origin. It is predominantly the scattering of a group of landslides across a discrete area, in close proximity to a fault, and their synchronous age rendering it likely that they were triggered by an earthquake (e.g., Jibson, 1996; Ojala et al., 2019). The spatial distribution of the two RSF deposits and the corresponding ages are therefore not conclusive regarding whether they were triggered by an earthquake or by other triggers, such as glacier debuttressing after deglaciation. However, the absence of a larger group of landslides could hint towards the nature of the Pärvie Fault rupture. It is in stark contrast to the large groups of earthquake-induced landslides nearby glacially-induced faults in northern Finland (e.g., Ojala et al., 2019). The presence of fault scarps but absence of landslides could support the occurrence of earthquakes underneath the retreating ice sheet. The cross-cut shorelines of IDLT indicate that the fault scarps locally ruptured at a close distance to the retreating ice margin. Although there is mounting evidence that the Pärvie Fault was not the result of a single rupture, it cannot be ruled out that there was a partial subglacial rupture. Sutinen et al. (2019) suggests morphological signs of subglacial rupture could be anastomosing networks of eskers (Fig. 10b) and subglacial crevasse fillings, which all seem to be present in the Torneträsk area (Ploeg, 2022).

## 5.5   Comparison to previously published maps and studies

Previously published work on glacial geomorphology in the Torneträsk Basin are broadly consistent with the results of this study but lack the detail allowed by the use of the LiDAR-based DEM. The most detailed geomorphological maps of the Torneträsk region were produced by Melander (1977a, b). His landform interpretation is based on aerial photographs and extensive field verification, and resulted in a comprehensive geomorphological map presented at 1:250,000. Our mapping (Fig. S1) is consistent with his mapping but adds considerable detail in terms of the number of raised shorelines (resulting in more ice-dammed lake stages), and the number of channels in flights of lateral meltwater channels. Additionally, whereas we map different types of meltwater channels, Melander (1977a, b) only categorizes glaciofluvial channels by size. For example, some large glaciofluvial channels correspond to outlet channels of ice-dammed lakes in this study. A critical difference between our maps is the number of lineations; our mapping includes significantly more lineations in both the premontane and the montane regions. The last glacial geomorphological map covering the Torneträsk region was produced from aerial photographs by Hättestrand (1998) at 1:1,250,000. Unlike Melander (1977a, b), this map includes large and small scale lineations, ribbed moraine, DeGeer moraines, and Veiki moraine. Our landform distributions of those features are consistent with the Hättestrand (1998) map but provide more detail, as individual lineations are outlined rather than a representative for a larger area. Thus, our mapping based on high-resolution LiDAR data, as expected, adds more detail in terms of landform count but is consistent with previously-mapped landform distributions. The critical implication of added detail in our mapping resides in a more detailed reconstruction of the ice-dammed lakes, but does not alter general inferences on ice retreat from ice flow directional indicators.

A LiDAR-based reconstruction of the ice-dammed lakes in the Torneträsk Basin resulted in eight stages, of which the two highest stages (T1 and T2) and the lowest stage (T8) were the least clear in their expression. Not surprisingly, therefore, Melander (1977c) identified at least five ice-dammed lake stages, which overlap in elevation with the stages T3–T7. It has been debated whether the shorelines reflect the presence of open lakes or ice-marginal lakes (Sjögren, 1908; Holdar, 1952; Melander, 1977c). The presence of shorelines on either side of the lake, with consistent elevations along the lake for the

different ice-dammed lake stages, strongly supports the notion of open lake systems for IDLT stages T3 to T8, while an absence of shorelines on southern valley slopes for IDLT stages T1 and T2 supports the existence of smaller ice-marginal lakes at that time. The notion of Sördalen and its canyon as an outlet for ice-dammed lakes in Torneträsk finds strong support in the literature (Sjögren, 1908; Holdar, 1952; Melander, 1977c). Melander (1977c) additionally suspected two potential outlets at the southeastern end of IDLT, but as the shorelines could not be traced to these proposed outlets, subglacial drainage is

mentioned as an alternative. Neither Sjögren (1908) nor Melander (1977c) mapped shorelines farther south than Jiekajärvi (not shown), while the southernmost shoreline in this study occurs 25 km farther south (than Jiekajärvi) at Alanen Kallovaara along Torneälven (Fig. 4d). Hence, this reconstruction shows that the lowest level of IDLT was more extensive than previously thought. Furthermore, an additional seven outlet channels could be identified and connected to the IDLT stages.

       Literature on outburst floods in this region of northern Sweden is lacking (Panin et al., 2020; Lützow et al., 2023), while

regions farther to the south have received more research attention (e.g., Elfström, 1987; Regnéll et al., 2019). The geomorphic traces of the GLOFs of IDLT stages T6–T8, which terminated in Ancylus Lake, were not described before. Although the Pärvie Fault has been the subject of much investigation (Lundqvist and Lagerbäck, 1976), the cross-cutting relationship between the Pärvie Fault and the oldest raised shorelines in the Torneträsk Basin has only become evident thanks to a regional analysis of shoreline gradients facilitated by recently released LiDAR data (Fig 6).

**6  Conclusions**

       A detailed reconstruction of spatial and temporal retreat patterns of the Fennoscandian Ice Sheet in the Torneträsk region of northwestern Sweden was predominately facilitated by mapping ice-dammed lake traces including raised shorelines, perched deltas, spillways, and outlet channels. Eight distinct ice-dammed lake stages were identified for the Torneträsk Basin, of which the lowest stages demonstrate the lake covered a larger extent than previously thought. Each stage had a different outlet, but

the lowest three converged in Tornedalen which saw multiple glacial lake outburst floods modifying its valley morphology and depositing deltaic flood deposits at the highest coastline of Ancylus Lake in the Baltic Sea Basin at c. 10 cal ka BP.

       The Pärvie Fault offsets the oldest six ice-dammed lake stages of Torneträsk, while the two youngest ice-dammed lake stages are not, highlighting future opportunities to directly date fault activity. Cross-cutting relationships between glacial landforms and fault scarps indicate that the Pärvie Fault ruptured multiple times during the last deglaciation, and close to the retreating

ice margin.

       Collectively, the glacial landforms, ice-dammed lake traces, and glacially-induced fault segments provide insight and detail into successive ice-marginal positions during deglaciation. While the ice sheet margin retreated south-southwestwards in a

relatively orderly fashion in the premontane region, it disintegrated into several ice lobes with intervening ice-free upland terrain in the montane region. Evidently, the topographic control on ice sheet retreat was significant. However, other factors played an important role too, such as the interaction between the ice margin and the lakes it dammed. Our reconstruction qualitatively supports previous reconstructions that the Torneträsk Basin became deglaciated around 9.9 cal ka BP, and the total duration of the eight ice-dammed lake Torneträsk stages is estimated to have been less than 200 yrs.

*Data availability.* The GIS shapefiles of the glacial landforms and a high resolution pdf-version of the glacial geomorphological map (Fig. S1) will be uploaded to an appropriate repository once the manuscript is accepted.

*Author contributions.* KP: Investigation, Analysis, Visualization, Writing - original draft preparation. APS: Conceptualization, Methodology, Supervision, Writing - review & editing.

*Competing interests.* Stroeven is member of the editorial boards of The Cryosphere, Solid Earth, and Earth Surface Dynamics.

*Acknowledgements.* We would like to thank the Swedish Polar Research Secretariat and SITES for the support of the work done at the Abisko Scientific Research Station, the Department of Physical Geography at Stockholm University for financially supporting travel, logistics, and equipment, Carl Regnéll for support and inspiring discussions, Gustaf Peterson Becher for providing and helping with the Python script for lake volume calculations, Robin Blomdin, Clas Hättestrand, Johan Kleman, Jan Mangerud, and Olle Melander for insightful feedback, and Colby Smith, Benjamin Boyes, and editor Caroline Clason for their thoughtful reviews that improved the article.

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
