# Peer review of "History and dynamics of Fennoscandian Ice Sheet retreat, contemporary ice-dammed lake evolution, and faulting in the Torneträsk area, northwestern Sweden"

_EGUsphere, 2024_

## Referee Comment (RC1)

[referee-annotated manuscript omitted]

---

## Author Comment (AC1)

*We thank Colby Smith for his thorough review of our work and, in responding to raised issues, we will be able to present an updated version of the manuscript that contains important improvements in its writing, arguments, and figures. Below we will address the reviewer comments in the order they were presented. The reviewer's comments are given in default font, with our original text, on which the comments were made, repeated in italics for reference. Our response is given in blue-, and suggested changes to the manuscript in red-type fonts.*

**RC1 – Colby Smith**

**General comments**
The manuscript is well organized around the geomorphic map, and draws conclusions related to the timing and dynamics of both the ice sheet retreat and the associated glacially induced faulting in northwestern Sweden. I recommend that the manuscript be published after minor revisions.

I have made some minor comments related to language, grammar, or general readability. These are included in the attached pdf of the manuscript.

All linguistic changes when adopted clearly improve the general readability of the manuscript. We would like to express our thanks for the reviewers' thorough check of the language.

Here, I will focus on the single larger issue that I believe needs to be addressed prior to publication. The authors make a case that different segments of the Pärvie fault ruptured at different times during or shortly after the late Weichselian deglaciation. Along Torneträsk, the authors present a strong case for fault rupture shortly after deglaciation based on the fact that older shorelines are displaced by the fault but not younger shorelines. Farther south, the other examples of where faulting either precedes or follows deglaciation are not as well presented. I do not question your conclusions, but that is because I have spent a lot of time looking at these faults in LiDAR. Other readers may need further convincing.

The crosscutting relationships are not obvious in figure 10 and they are not adequately described in the text. Thus, I suggest revising figure 10 to be more like figure 3 and clearly show the crosscutting relationships between the fault and glacial landforms, and the fault and the shorelines. Additionally, include text that explains the crosscutting relationships and the conclusions drawn from them.
This is a nice contribution to the deglacial history of northern Sweden, and I look forward to seeing it published.

First, we want to thank the reviewer for his kind words and encouragement. We are pleased that our work is considered a contribution to the deglacial history of northern Sweden. Second, we thank the reviewer for pointing out the issue regarding the crosscutting relationships and for giving us the opportunity to revise Figure 10. We will provide a detailed response and revisions further down in the document, where we present an improved Figure 10 and a suggestion for an additional panel to Figure 6.

**Specific comments (as annotated in PDF)**

- *Lines 74-76: "Finally, a refined history of ice-marginal retreat potentially enables future investigations of the interaction between the retreating ice sheet and the re-activation of faults through glacial isostatic adjustment (Fig. 1b)."*

The primary stress related to these faults is tectonic (ie related to the spreading of the Atlantic). Glacial isostatic adjustment is more the trigger than the cause. See: Stewart, I.S., Sauber, J. & Rose, J., 2000: Glacio-seismotectonics: ice sheets, crustal deformation and seismicity, Quaternary ScienceReviews 19, 1367–1389.

The sentence will be adjusted to emphasize how the total stress field is comprised of both the presence of a regional tectonic stress and the superimposed glacially-induced stress.

"Finally, a refined history of ice-marginal retreat potentially enables future investigations of the interaction between the changing configuration of the retreating ice sheet and its marginal positions and a re-activation of faults through the overprinting of the prevailing regional tectonic stress with glacially-induced stress".

- *Line 80: "The Torneträsk Basin cuts through the Scandinavian mountain range, also known as the Scandes, and across the border to Norway forms the headwaters of Rombaksfjorden (Rombaken; Fig. 2)."*

There is no subject in this clause. It is incomplete. What forms the headwaters? It's also unclear because it sounds like the Torneträsk basin is the headwaters of the fjord. Torneträsk drains to the east, the fjord is over a pass to the west.

The sentence will be altered to emphasize that Torneträsk basin is draining to the east. The confusion probably stemmed from Fig. 2 showing that during deglaciation the Fennoscandian Ice Sheet drained westwards across the Torneträsk basin and across the water divide, providing meltwater to Rombaksfjorden.

"The Torneträsk Basin cuts through the Scandinavian mountain range, also known as the Scandes, and drains to the east. Across the border to Norway, over a pass to the west, the headwaters of Rombaksfjorden are found (Rombaken; Fig. 2)."

- *Line 119: "Additional azimuths of 90°, °and 180°"...*

Should there be another direction here?

Thank you for finding this mistake! The degree symbol will be removed, there was no other azimuth used than the two mentioned.

"Additional azimuths of 90° and 180°"

- *Line 259: "... there are no other geomorphological cross-cutting relationships that show the exact offset as well as the raised shorelines."*

This is true, but the crosscutting relationship between fault and shoreline is not visible in the LiDAR imagery.

We suspect the confusion stems from the use of the phrase "as well as", which was meant as "and also", not as "as good as". The crosscutting relationship between the fault and shorelines is only evident from regional analyses using the graph that plots the shoreline elevations along a reference plane (old and updated Figure 6, below), not from the LiDAR imagery itself due to the lack of continuity of the shorelines at the location of the fault scarp. This study is basically outlining a new technique to identify fault ruptures. The sentence will be changed to emphasize that the crosscutting is not visible in the LiDAR imagery.

"… there are no geomorphological cross-cutting relationships visible in the LiDAR imagery that show the offset of raised shorelines at the exact location of the fault scarp (Figure 6)."

Old Figure 6:

[Figure]

New Figure 6:

[Figure]

Figure 6. (a) Lake stages identified from the elevations of raised shorelines, perched deltas, and outlet channels of ice-dammed lake Torneträsk. At the Abscissa value of zero, the ordinate value is 342 m a.s.l., the current elevation of the surface of Torneträsk. The approximate location where the Pärvie Fault crosscuts the Torneträsk Basin is indicated by the red bar. The distance is calculated along an axis perpendicular to the isobases of postglacial rebound of the shorelines (see Fig. 1c). The corresponding elevation ranges are summarised in Table 2. (b) Elevations of raised shorelines of ice-dammed lake Torneträsk on either side of the Pärvie Fault where it crosscuts the northern shore of Torneträsk (see red bar in (a)), illustrating elevation jumps of around 8 m for the higher raised shorelines (T3-T6), while the lowest raised shoreline (T7) crosses the fault at 365–366 m a.s.l. The background is a shaded relief based on the DEM provided by ©Lantmäteriet.

- *Lines 260-261: "Remarkably, ice-dammed lakes T1 and T2 do not appear to have been offset by the Pärvie Fault, although this has perhaps remained unrecorded due to a lack of shoreline segments for these lake stages."*

I would revise this to indicate that these older shorelines do not record the fault displacement because of the lower resolution of the data. If there were continuous shorelines they would be crosscut by the fault.

We agree that the shorelines would be crosscut by the fault if they were continuous. The sentence will be adjusted to emphasize this argument.

"Ice-dammed lakes T1 and T2 do not record the offset by the Pärvie Fault due to the lower resolution of the data (Fig. 6)."

- *Line 283-284: "Mapping of surficial geology by the Geological Survey of Sweden (SGU, 2024a) shows that" …*

This is technically cited correctly if it came from the database this year. It is, however, significant that the mapping was carried out on aerial photos not lidar (in the 1990s I think).

First, while working on this comment, we concluded that the in-text citation refers mistakenly to the wrong SGU dataset (2024a instead of 2024b). Second, the original URL of SGU (2024b) links to a pdf with a product description of a surficial geology dataset at another scale. Both these errors will be corrected in the text and in the reference list.

"SGU: Product description: Surficial geology 1:250 000, northernmost Sweden (Swedish), https://resource.sgu.se/dokument/produkter/ jordarter-250000-nordligaste-sverige-beskrivning.pdf, 2024b."

We agree that the reference could be misleading to infer that the work in it was performed recently. However, it is correct to cite a product description from SGU which was last updated in 2024. Its maps are based on compilation and digitization of older surveys (the document does not clarify from which years), where the mapping was indeed mainly based on aerial photo interpretation together with field observations along the sparse road network. Elevation models were not used. The compilation was finished in 2011. In a separate guide by SGU to the surficial geology maps and databases of Sweden, it appears they started to use aerial imagery for mapping in the 1980s. In the digital map viewer, it shows that most of the mapping was finished in early 2000s.

"Mapping of surficial geology by the Geological Survey of Sweden (SGU, 2024b), carried out from the 1980s to early 2000s based on aerial imagery supported by sparse field observations, shows that vast amounts of glacio-fluvial sediments which form deltaic deposits are found 165 km downstream of the initial drainage location(s) along both Tornedalen and Kalixdalen (Fig. 8d)."

- *Lines 379-382: "These are favorable comparisons because GLOFs from Akkajaure and Sitojaure cut the Ancylus Lake highest coastline and are therefore in timing close to the youngest GLOF of Torneträsk (and so are their shoreline gradients) and lake evolution in central Jämtland spans 10.5–9.2 cal ka BP (Regnéll et al., 2023), which brackets ice retreat from the Torneträsk Basin, rendering it reasonable that the gradients of IDLT fall within the range of values from central Jämtland."*

This is too much information for one sentence. Break this up into smaller sentences and move the parenthetical text into the body of a sentence.

We appreciate the opportunity to revise this section. The study from Regnéll et al. (2019) did not trace the GLOFs from Akkajaure and Sitojaure (the two lakes we cited), but Pieskehaure and Mavasjaure, which are approx. 90 km farther south. These two ice-dammed lakes were, however, not reconstructed, so there are no shoreline gradients available to compare to. We will therefore remove this statement from our argumentation. Breaking up the sentence as the reviewer suggested will improve the readability as well.

"These are favorable comparisons because ice-dammed lakes Akkajaure and Sitojaure also formed in response to the final deglaciation of the Fennoscandian Ice Sheet. Their timing is closest to the youngest GLOF of Torneträsk, and so are their shoreline gradients. Additionally, lake evolution in central Jämtland spans 10.5–9.2 cal ka BP (Regnéll et al., 2023), which brackets ice retreat from the Torneträsk Basin, rendering it reasonable that the gradients of IDLT fall within the range of values from central Jämtland."

- *Line 386: "The reconstruction of the ice-dammed lakes in the Torneträsk Basin using LiDAR resulted in eight stages"* …

Add a sentence at the beginning of this paragraph that allows the reader to know where you headed. "Previously published work on ice-dammed lakes in the Torneträsk basin are broadly consistent with the results of this study but lack the detail allowed by use of the DEM."

Thank you for the helpful suggestion.

"Previously published work on ice-dammed lakes in the Torneträsk Basin are broadly consistent with the results of this study but lack the detail allowed by the use of the LiDAR-based DEM".

- *Lines 432-435: "The amplifying effect that ice-marginal lakes have on retreat rates of lacustrine-terminating ice sheets (e.g., Stokes and Clark, 2004; Utting and Atkinson, 2019), explains that the FIS experienced higher ice losses due to the ice-dammed lakes in the Torneträsk region, and helps explaining the pivoting motion of ice retreat in this region."*

Flip this sentence around. "Higher ice losses due to the ice-dammed lakes in the Torneträsk region can be explained by the amplifying effect…… Thus, the presence of the ice-dammed lakes led in part to the pivoting…"

This suggestion, when adopted, indeed helps improve readability.

"Higher ice losses of the FIS due to the ice-dammed lakes in the Torneträsk region can be explained by the amplifying effect that ice-marginal lakes have on retreat rates of lacustrine-terminating ice sheets (e.g., Stokes and Clark, 2004; Utting and Atkinson, 2019). Thus, the presence of the ice-dammed lakes led in part to the pivoting motion of ice retreat in this region."

- *Line 437-438: "The ice-marginal positions that dammed the successive ice-dammed lake stages of Torneträsk fall approximately in-between their 10.1 and >9.9 cal ka BP isochrons (Fig. 1b)"* …

How many dates actually constrain these isochrons?

Few, if any; there is a real dearth of data in this region. See Hughes et al. (2016) and Stroeven et al. (2016) for data compilations and resulting deglaciation histories.

- *Lines 460-461: "In fact, the fault may have ruptured subglacially as well while as much as 95 km of the fault trace was ice covered at that point."*

If the fault ruptured along its entire length at this time, then as much as 95 km.......

Thank you for pointing this out, because it is indeed questionable whether the fault ruptured along its entire length.

"In fact, if the fault ruptured along its entire length at this time, then as much as 95 km of the fault trace were ice covered at that point."

- *Line 466: ... "and pre-date deglaciation south of this ice sheet margin (Fig. 10)."*

As originally suggested by Lundqvist and Lagerbäck (1976).

Citing this paper highlights how the findings of our study are in contrast to the original school of thought.

"If all of the Pärvie Fault ruptured at once, such as is typically considered when calculating the amount of energy released, cross-cutting should post-date deglaciation north of the inferred ice sheet margin at the time of rupture (between T6 and T7, Figs. 9e and 9f) and pre-date deglaciation south of this ice sheet margin, as originally suggested by Lundqvist and Lagerbäck (1976) (Fig. 10)."

- *Figure 10*

I can't see these crosscutting relationships in the images provided. I suggest a revised figure similar to Fig. 3 with larger, clearer lidar images (perhaps without slope?). Additionally, to demonstrate the uncertainty involved here, include a lidar image of your crosscut shorelines. You present a convincing case that the shorelines are cut by the fault, but I can't see it in the imagery alone. This should be presented and discussed in the text as well.

We thank the reviewer for giving us the opportunity to revise Figure 10 to the standard of Figure 3. We have followed his suggestion, almost entirely. Below the former Figure 10 map, we now indeed will present four panels with significantly improved LiDAR imagery showing the cross-cutting relationships referenced in the manuscript. The four panels show one example of the Pärvie fault postdating (cutting) the glacial landforms or postglacial fluvial landforms, one instance where an esker drapes the fault trace, thus indicating faulting before esker formation, and two panels where we tentatively conclude that fluvial terraces have been offset multiple times. What is not included is a panel that shows the Pärvie fault cutting the shorelines and off-setting them on either side of the fault trace. If this evidence existed, we suspect this information might have been presented earlier-on. Rather, it is the painstaking reconstruction of the shoreline fault traces along the full length of the Torneträsk basin and across the full range of elevations that shows this jump in elevation of the oldest shorelines at the location of the Pärvie Fault. We include an additional map in our response (a panel that could be presented together with former Fig. 6) which illustrates the shoreline traces that are visible in the immediate surroundings of the Pärvie Fault trace (<1 km) and the elevation jumps (if any) that can be gleaned from these. Including this figure clarifies the question by the reviewer, but does perhaps

not contain enough supportive information to warrant inclusion? Would the editor like to advise us on whether inclusion of this figure would be required, appreciated, or discouraged?

We believe that improved Figure 10 and its informative caption covers the information sought by the reviewer, and we abstain from further inclusion of a discussion paragraph as the imagery speaks the language and because this information aligns with excellent papers written by the reviewer himself (Smith et al., 2018, 2021), to which we direct the readers if they are interested in the Pärvie Fault.

Old Figure 10:

[Figure]

New Figure 10:

[Figure]

Figure 10. (a) Observed cross-cutting relationships between the glacial geomorphology and the fault scarp traces of the Pärvie Fault. Examples include cross-cutting (b) pre-dating deglaciation, where an esker drapes a fault scarp, (c) post-dating deglaciation, where glaciofluvial landforms are cut by a fault scarp, and (d-e) occurring, tentatively, multiple times, where fluvial terraces are offset by multiple ruptures (panel (d) is the same location as in Smith et al. 2021, Fig 12.4). The background is a shaded relief based on the DEM provided by ©Lantmäteriet.

New panel (b) in Figure 6:

[Figure]

Figure 6. (a) Lake stages identified from the elevations of raised shorelines, perched deltas, and outlet channels of ice-dammed lake Torneträsk. At the Abscissa value of zero, the ordinate value is 342 m a.s.l., the current elevation of the surface of Torneträsk. The approximate location where the Pärvie Fault crosscuts the Torneträsk Basin is indicated by the red bar. The distance is calculated along an axis perpendicular to the isobases of postglacial rebound of the shorelines (see Fig. 1c). The corresponding elevation ranges are summarised in Table 2. (b) Elevations of raised shorelines of ice-dammed lake Torneträsk on either side of the Pärvie Fault where it crosscuts the northern shore of Torneträsk (see red bar in (a)), illustrating elevation jumps of around 8 m for the higher raised shorelines (T3-T6), while the lowest raised shoreline (T7) crosses the fault at 365–366 m a.s.l. The background is a shaded relief based on the DEM provided by ©Lantmäteriet.

- Line 480: "Currently, such an approach appears unrealistic given the mounting evidence for different types of cross-cutting relationships (Fig. 10), reinforcing observations by Lundqvist and Lagerbäck (1976) that the Pärvie Fault ruptured both prior to, and after, deglaciation."

This was published before people knew that there were multiple generations of glacial landforms in N. Sweden. Thus, their conclusion was that a single rupture occurred partially under the ice.

Thank you for this reminder. To avoid confusion, this citation will be removed together with the statement about the rupture being both prior and after deglaciation. Instead, we will add a sentence emphasizing how the cross-cutting evidence reinforces observations of multiple ruptures.

"Currently, such an approach appears unrealistic given the mounting evidence for different types of cross-cutting relationships (Fig. 10), reinforcing observations that the Pärvie Fault did not rupture at once."

- *Line 535-536: "Cross-cutting relationships between glacial landforms and fault scarps indicate that the Pärvie Fault ruptured multiple times during the last deglaciation and, indeed, before the last deglaciation."*

What do you mean by this? Before the last glaciation was complete?
You do not discuss evidence of pre-late Weichselian faulting.

Indeed, we have no evidence for pre-late Weichselian faulting. We only have evidence of faulting during deglaciation, where the rupture happened shortly after ice retreat. We will remove the latter part of the sentence that insinuated rupture before the last deglaciation.

"Cross-cutting relationships between glacial landforms and fault scarps indicate that the Pärvie Fault ruptured multiple times during the last deglaciation."

**References**

Regnéll, C., Mangerud, J., & Svendsen, J. I. (2019). Tracing the last remnants of the Scandinavian Ice Sheet: Ice-dammed lakes and a catastrophic outburst flood in northern Sweden. *Quaternary Science Reviews, 221*. https://doi.org/10.1016/j.quascirev.2019.105862

Smith, C. A., Grigull, S., & Mikko, H. (2018). Geomorphic evidence of multiple surface ruptures of the Merasjärvi "postglacial fault", northern Sweden. *GFF, 140*(4), 318–322. https://doi.org/10.1080/11035897.2018.1492963

Smith, C. A., Mikko, H., & Grigull, S. (2021). Glacially Induced Faults in Sweden: The Rise and Reassessment of the Single-Rupture Hypothesis. In H. Steffen, O. Olesen, & R. Sutinen (Eds.), *Glacially-Triggered Faulting* (pp. 218–230). Cambridge University Press. https://doi.org/10.1017/9781108779906.016

---

## Author Comment (AC2)

*We thank Benjamin Boyes for his thorough review of our work and, in responding to raised issues, we will be able to present an updated version of the manuscript that contains important improvements in its writing, arguments, and figures. Below we will address the reviewer comments in the order they were presented. The reviewer's comments are given in default font, with our original text, on which the comments were made, repeated in italics for reference. Our response is given in blue-, and suggested changes to the manuscript in red-type fonts.*

**RC2 – Benjamin Boyes**

**General remarks**

This article presents a new framework for understanding late glacial landscape evolution in northern Sweden. The study uses original geomorphological data and previously published chronometric data to reconstruct Fennoscandian Ice Sheet retreat patterns, ice dammed lake development, and the evolution of post-glacial faults. This publication is suitable for publication in *The Cryosphere* after minor revisions, and I look forward to seeing it published.

We thank the reviewer for his encouraging feedback and for considering our work a valuable contribution to the understanding of late glacial landscape evolution in northern Sweden. We are pleased that the reviewer finds our study fit for publication.

*Academic rigour and accuracy*
The study's methodology is comprehensive, and the results are clearly laid out. However, it would be useful if the following points are clarified:

- The mapping methods could be more clearly laid out. The manuscript suggests you mapped a wide array of features, but details (e.g. how polylines are drawn to map landforms) are only provided for ice dammed lakes (and associated features).

The mapping approach for the other landforms will be described in an extra column in Table 1, while the more comprehensive description for the ice-dammed lake traces is kept in the text as it was. We will add a sentence referring to Table 1 for the details on the mapping approach.

"The mapping approach, that is, how the landforms are delimited in GIS software, is briefly described for all landforms in Table 1. Given the focus on ice-dammed lake traces, the mapping approach of raised shorelines and perched deltas, and the methodology to identify ice-dammed lake stages, are described in more detail below."

- The fault lines and rock slope failure deposits are not presented in the results. These data are from previous work (as suggested by Figure 1b) and the source of these data need to be more obviously discussed in the text. If you checked these against the LiDAR data, this needs to be discussed.

Thank you for this suggestion. We will add a paragraph in the Methods/Datasets section to discuss the data sources in more detail.

"Vector datasets of previously published studies were used for different purposes. The international database of Munier et al. (2020) contains glacially-induced faults in northern Fennoscandia (Fig. 1b), of which many were previously proposed and recently confirmed based

on the recent LiDAR data. The faults in the database were cross-referenced with the LiDAR-based DEM, but no effort was made to identify new faults. The dataset was used to identify cross-cutting relationships between glacial landforms and fault scarps. The deglaciation isochrons reconstructed by Stroeven et al. (2016) were used to evaluate the implications of the direction of mapped landforms and to constrain the chronology (Fig. 1b). Cosmogenic nuclide [10]Be exposure ages of two rock slope failure (RSF) deposits were taken from Stroeven et al. (2002, 2016). The RSF extents were cross-referenced against the LiDAR-based DEM."

- The mapping is good and has added considerable detail to the region, and I like how clear the supplementary map is. However, from personal experience mapping landforms in this region from similar LiDAR data and in the field, I think some features have not been mapped. This is entirely subjective, but it would be good to know why you chose to map certain features and if you chose to omit any?

We are not sure whether this comment is referring to entire feature classes or to individual features: in our answer we presume that latter. Although there is of course the aim to identify all features, there were certainly features where the actual landform type remained ambiguous. In this respect the map is conservative: we only mapped features of which the genetic interpretation could be confidently determined. Given that dataset, we were able to draw robust conclusions, which are insensitive to the total number of mapped features.

- The relative timing (e.g. last glacial vs previous glacial) of some of the features needs clarification. Why have you decided which landforms are pre-last glacial, and maybe show these features on a map of their own? You mention that this is a thing, but don't provide any evidence of pre-last glacial landforms.

We appreciate the reviewer's comment regarding the relative timing of our mapped features. We understand the importance of distinguishing between last glacial and pre-last glacial landforms. However, we do not believe that we explicitly stated it as a significant issue ("a thing") in our manuscript. Our intention was to provide a general context, as we did in the following sections:

- **Study area section (lines 95-103):** We describe that the area is known for its palimpsest landforms.
- **Discussion/Ice-marginal positions section (lines 411-413):** We reiterate that it is known that landforms are not exclusively from the last deglaciation.
- **Discussion/Glacially-induced faulting section (lines 470-475):** We mention that traces of older glaciations can complicate reconstructions, such as those of relative timing between fault rupture and glacial landforms. We mention that glacial landforms cut by the Pärvie Fault within the study area align with reconstructed deglaciation directions.

We are uncertain about the improvements the reviewer is suggesting. After careful consideration, we believe that the current presentation aligns best with the overall objectives of our manuscript. Therefore, we will retain the original content on this topic. We thank the reviewer for making us re-think this topic.

- As I understand it, mkm$^{-1}$ is a unit referring to slope? A short sentence clarifying what this means would be helpful.

Thank you for pointing this out: it is indeed a unit referring to the gradient of the shorelines. A sentence will be added to the Methods chapter to explain the unit.

"The tilting of the shorelines is described as a gradient in m km$^{-1}$ where the elevation difference (in meters) is given over the distance (in kilometers) in direction of the reference plane."

- Some discussion on how your geomorphological mapping compares with existing geomorphological maps could be interesting. You do this comprehensively for the lakes, but not the other landforms.

This is a good idea. It requires two steps. First, we need to explain what data sources we have for comparison and how they were digitized, and then we will suggest a section discussing how our mapping compares to the previously published maps.

A statement on the inclusion of printed maps for cross-referencing will be included in the Methods chapter.

"Several printed maps were digitized to cross-reference the mapping. The geomorphological maps by Melander (1977a, 1977b) were georectified by the Agency for Digital Government (DIGG; https://www.digg.se/en), which were then georeferenced in GIS software using locations on the map with known coordinates. Additionally, a scanned and georeferenced map by Hättestrand (1998) was imported into the GIS environment."

We will add another paragraph to our Discussion where we will present a global comparison between our mapping and previous maps.

"The most detailed geomorphological maps of the Torneträsk region were produced by Melander (1977a, 1977b). His landform interpretation is based on aerial photographs and extensive field verification, and resulted in a comprehensive geomorphological map presented at 1:250,000. Our mapping (Fig. S1) is consistent with his mapping but adds considerable detail in terms of the number of raised shorelines (resulting in more ice-dammed lake stages), and the number of channels in flights of lateral meltwater channels. Additionally, whereas we map different types of meltwater channels, Melander (1977a, 1977b) only categorizes glaciofluvial channels by size. For example, some large glaciofluvial channels correspond to outlet channels of ice-dammed lakes in this study. A critical difference between our maps is the number of lineations; our mapping includes significantly more lineations in both the premontane and the montane regions. The last glacial geomorphological map covering the Torneträsk region was produced from aerial photographs by Hättestrand (1998) at 1:1,250,000. Unlike Melander (1977a, b), this map includes large and small scale lineations, ribbed moraine, DeGeer moraines, and Veiki moraine. Our landform distributions of those features are consistent with the Hättestrand(1998) map but provide more detail, as individual lineations are outlined rather than a representative for a larger area. Thus, our mapping based on high-resolution LiDAR data, as expected, adds more detail in terms of landform count but is consistent with previously-mapped landform distributions. The critical implication of added detail in our mapping resides in a more detailed reconstruction of the ice-dammed lakes, but does not alter general inferences on ice retreat from ice flow directional indicators."

- Table 1: This is a nice comprehensive table – I particularly like the "possible identification error" column. It would be helpful to have in this table a column that explains the mapping approach, as an example figure in e.g. Boyes et al., 2021 (https://doi.org/10.1080/17445647.2021.1970036) and/or as text.

This suggestion will improve the description of our mapping approach. We will add the mapping approach as the last text column in Table 1.

[revised manuscript text omitted]

- Figure 1: In panel b, consider thinning out the isochrons or making the panel bigger. At present, it's a little difficult to see all of the components in the figure.

We enlarged panels a and b and thinned out the isochrons and country borders in panel b, which made it easier to appreciate all components in the figure.

Old Figure 1:

[Figure]

New Figure 1:

[Figure]

- Figure 9: If you are unsure whether the ice sheet also retreated into the Kebnekaise/Sarek Mountains, consider leaving a ? symbol over these locations in your retreat pattern to acknowledge this.

We will add another question mark in panel g over the mountains of Kebnekaise, which better visualises that the retreat at this location remains unconstrained. The Sarek Mountains are well outside the mapping boundaries.

- Figure 10: The cross-cutting is really difficult to see. Make the panels bigger with nice and clear LiDAR hillshade images.

We thank the reviewer for his comment, which has also been raised by reviewer 1. We present an improved Figure 10 below.

Old Figure 10:

[Figure]

New Figure 10:

[Figure]

Figure 10. (a) Observed cross-cutting relationships between the glacial geomorphology and the fault scarp traces of the Pärvie Fault. Examples include cross-cutting (b) pre-dating deglaciation, where an esker drapes a fault scarp, (c) post-dating deglaciation, where glaciofluvial landforms are cut by a fault scarp, and (d-e) occurring, tentatively, multiple times, where fluvial terraces are offset by multiple ruptures (panel (d) is the same location as in Smith et al. 2021, Fig 12.4). The background is a shaded relief based on the DEM provided by ©Lantmäteriet.

- *Line 119: ... "as these are considered the optimal values for the visualization of hillshade relief models for the purpose of glacial geomorphological mapping (Chandler et al., 2018)."*

Chandler et al., 2018 don't suggest these values for hillshade images, other authors do (specifically Chandler cite Smith and Clark, 2005 and Hughes et al., 2010). Change (or add) the citation to other sources.

Thank you for spotting that, the older citations will be adopted.

"The DEM was processed in ArcGIS Pro 2.9.3. to create a hillshade relief model using an illumination angle with an altitude of 30° and azimuths of 45° and 315°, as these are considered the optimal values for the visualization of hillshade relief models for the purpose of glacial geomorphological mapping (Smith and Clark, 2005; Hughes et al., 2010)."

- *Lines 119-121: "Additional azimuths of 90°, ° and 180°, perpendicular and parallel to the dominant lineation orientation, respectively, were applied to reduce the 'azimuth bias' (Smith and Clark, 2005; Chandler et al., 2018)."*

Either remove the statement or define which azimuths were used.

The degree symbol has been removed, there was no other azimuth used than the two already mentioned. Reviewer 1 had the same comment, thank you for spotting this.

"Additional azimuths of 90° and 180°"

- *Lines 164-167: "The glacial geomorphology of the Torneträsk Basin is presented in Fig. S1. The total comes to 6633 mapped features, of which there are 2796 lineations, 678 eskers, 39 ribbed moraine, 1262 meltwater channels, 155 marginal moraines, 510 undifferentiated moraines, 894 raised shorelines, 206 perched deltas, 25 outlet channels, and 38 veiki moraines. Note that the count includes all segments of a landform, so it represents a feature count instead of a landform count."*

Here you provide numbers of how many features you have mapped. However, because you have not detailed the mapping approach for each landform type, it is not clear whether the quoted 6,633 mapped features are individual features or groups of features. For example, you say you have mapped 38 veiki moraines – is that 38 areas of veiki moraine, or 38 individual veiki moraine plateau?

The mapping approach is now described in Table 1, which clarifies that ribbed moraine and veiki moraine are mapped as areas, rather than individual ridges or plateaus. The text will also be changed to emphasize that the feature count refers to the number of ribbed moraine and veiki moraine areas.

"The total comes to 6633 mapped features, of which there are 2796 lineations, 678 eskers, 39 areas of ribbed moraine, 1262 meltwater channels, 155 marginal moraines, 510 undifferentiated moraines, 894 raised shorelines, 206 perched deltas, 25 outlet channels, and 38 areas of veiki moraine."

Later on in the results section, you go on to say landforms are found "relatively often" in x locations. It would be better to put a number (i.e. %) on this.

We will specify the percentage of landforms where they were mentioned in relation to the montane and the premontane regions, in their respective Results sections.

Line 170:
 "Lineations occur across the area but are most common in the premontane region (60%, Fig. 3a)."

Line 179:
"Eskers occur across the area, but are most frequent in the montane region (63%)."

Lines 187-188:
"Subglacial meltwater channels are prevalent in the entire study area, although most of them occur in the premontane region (72%)."

Lines 193-194:
"Ribbed moraine occurs predominantly in the premontane region (56%) and in the montane region on uplands north of Torneträsk and in between Rautasjaure and Torneträsk (Fig. S1)."

Line 204:
"Whereas subglacial channels are abundant in the premontane region (72%), lateral meltwater channels are relatively rare (21%)."

Line 223:
"Moraines are virtually lacking in the premontane region (4%, Fig. S1)" …

- *Lines 257-259: "However, whereas there is abundant information on fault displacement of glacial geomorphology, indicating that the Pärvie Fault ruptured after landform formation (Figs. 3b and 3c), there are no other geomorphological cross-cutting relationships that show the exact offset as well as the raised shorelines."*

Please point to this cross-cutting relationship on the figure.

Thank you for asking clarification on this, reviewer 1 commented on the same lines. The crosscutting relationship between the fault and shorelines is only evident from regional analyses using the graph that plots the shoreline elevations along a reference plane, not from the LiDAR imagery itself due to the lack of continuity of the shorelines at the location of the fault scarp. This study is basically outlining a new technique to identify fault ruptures. Additionally, a new potential Figure 6 is presented to show the raised shorelines at the location where the fault crosscuts the basin.

"there are no geomorphological cross-cutting relationships visible in the LiDAR imagery that show the offset of raised shorelines at the exact location of the fault scarp."

Old Figure 6:

[Figure]

New Figure 6:

[Figure]

Figure 6. (a) Lake stages identified from the elevations of raised shorelines, perched deltas, and outlet channels of ice-dammed lake Torneträsk. At the Abscissa value of zero, the ordinate value is 342 m a.s.l., the current elevation of the surface of Torneträsk. The approximate location

where the Pärvie Fault crosscuts the Torneträsk Basin is indicated by the red bar. The distance is calculated along an axis perpendicular to the isobases of postglacial rebound of the shorelines (see Fig. 1c). The corresponding elevation ranges are summarised in Table 2. (b) Elevations of raised shorelines of ice-dammed lake Torneträsk on either side of the Pärvie Fault where it crosscuts the northern shore of Torneträsk (see red bar in (a)), illustrating elevation jumps of around 8 m for the higher raised shorelines (T3-T6), while the lowest raised shoreline (T7) crosses the fault at 365–366 m a.s.l. The background is a shaded relief based on the DEM provided by ©Lantmäteriet.

- *Lines 419-423: "Hence, a strong control of topography on ice retreat patterns and rates is evident, as other studies have demonstrated for the FIS (Stroeven et al., 2016; Szuman et al., 2024), the British-Irish Ice Sheet (Greenwood et al., 2007; Hughes et al., 2014), and the Cordilleran Ice Sheet (Kleman et al., 2010; Dulfer et al., 2022)."*

Topographic controls on ice sheet geometry during retreat of a thinning ice sheet have also been highlighted in northwest Russia (https://doi.org/10.1002/jqs.1130; https://doi.org/10.1016/j.quascirev.2022.107872; https://doi.org/10.1111/bor.12653).

Thank you for these suggestions. We will add another citation to represent the deglaciation of the northwestern sector of the Fennoscandian Ice Sheet.

"Hence, a strong control of topography on ice retreat patterns and rates is evident, as other studies have demonstrated for the FIS (Stroeven et al., 2016; Boyes et al., 2023; Szuman et al., 2024)" …

- *Lines 436-439: "In the reconstruction of Stroeven et al. (2016), the retreating ice margin swept across the study area in a time span of 500yr (Fig. 1b). The ice-marginal positions that dammed the successive ice-dammed lake stages of Torneträsk fall approximately in-between their 10.1 and >9.9 cal ka BP isochrons (Fig. 1b), which would suggest the ice-dammed lake system of Torneträsk existed for a total duration of <200 yr."*

You briefly mention timing of lakes here and have more detail on faulting chronology in Section 5.4. Could you have a single chronology section that deals with the chronologies of each component (ice sheet retreat, ice dammed lake formation/drainage, and faulting) as they are interlinked.

We agree that the chronologies of each component in this reconstruction are interlinked and that it is worth exploring whether a single chronology section improves the structure of the paper. We will aim to consolidate the chronologies of each component into a single section.

It would be better to use the point chronometric data presented by Stroeven et al., 2016 and in the DATED-1 database rather than comparing to the isochrons as this may provide more relevant information for your reconstruction.

The point chronometric data of Stroeven et al. (2016) and the DATED-1 database present challenges for comparison with our mapping due to limited constraints over a large area. Stroeven et al. (2016) lack landform types, making it difficult to draw any conclusions about ages without going into the geomorphological context of every sample individually. Many samples are from bedrock, representing cumulative exposure from previous ice-free periods.

The DATED-1 database includes only seven individual ages within our study area, of which four are deglacial. Three of these deglacial ages are radiocarbon dates from the same moraine-dammed lake, of which the location is stored incorrect in the database.

Given the scarcity of data in our region, we believe that a comparison with point chronometric data would not significantly enhance our reconstruction. We will therefore refrain from implementing the suggested changes.

- *Line 515: "There are two large rock slope failure (RSF) deposits in the study area that were potentially triggered by ruptures along the Pärvie Fault."*

You've suggested that the rock slope failure deposits are a result of post-glacial earthquakes. Such landslides can also be triggered by glacial de-buttressing during glacier retreat. You should include some discussion on this point, and if you still consider these landslides to be earthquake induced, then you need to clearly provide evidence for this.

We agree that this statement needs to be discussed, and we will add a paragraph in our Discussion, arguing that we cannot conclude whether the rock slope failure deposits are earthquake-induced or the product from other processes. The comment also inspired to look for more geomorphological evidence regarding subglacial fault rupture.

"The formation of landslides in northern Fennoscandia has been associated to earthquakes caused by post-glacial faulting (Sutinen, 2005; Lagerbäck and Sundh, 2008). There are two large landslides in the study area, but the absence of a larger group of landslides in the vicinity of the Pärvie Fault challenges the potential earthquake-induced origin. It is predominantly the scattering of a group of landslides across a discrete area, in close proximity to a fault, and their synchronous age rendering it likely that they were triggered by an earthquake (e.g., Jibson, 1996; Ojala et al., 2019). The spatial distribution of the two RSF deposits and the corresponding ages are therefore not enough evidence to conclude whether they were triggered by an earthquake or by other triggers, such as glacier debuttressing after deglaciation.

The absence of a group of landslides could hint towards the nature of the Pärvie Fault rupture. It is in stark contrast to the large groups of earthquake-induced landslides nearby glacially-induced faults in northern Finland (e.g., Ojala et al., 2019). The presence of fault scarps but absence of landslides could support the occurrence of earthquakes underneath the retreating ice sheet. The crosscut shorelines of Torneträsk indicate that the fault scarps locally ruptured at a close distance to the retreating ice margin. Although there is mounting evidence that the Pärvie Fault was not the result of a single rupture, it cannot be ruled out that there was a partial subglacial rupture. Sutinen et al. (2019) suggests morphological signs of subglacial rupture could be anastomosing networks of eskers (Fig. 10b) and subglacial crevasse fillings, which are both present in the Torneträsk area (Ploeg, 2022)."

---

## Author Response (AR1)

*We thank editor Caroline Clason, and reviewers Colby Smith and Benjamin Boyes, for their thorough reviews of our work and, in responding to raised issues, we are presenting an updated version of the manuscript that contains important improvements in its writing, arguments, and figures. Below we will address the comments in the order they were presented. The comments are given in default font, with our original text, on which the comments were made, repeated in italics for reference. Our response is given in blue-, and changes to the manuscript in red-type fonts.*

**Editor – Caroline Clason**

- *Lines 123–124: "Natural colour (RGB 4, 3, 2) and colour infrared (RGB 5, 4, 3) orthophotos with a resolution of 0.5 m were also used in the mapping project."*

The data is described well, although you could include the date(s) and source(s) of the orthophotos in section 3.1 when providing a revised version of the manuscript.

Orthophotos were provided by Lantmäteriet and were acquired in 2018 for roughly the montane region in the western part and in 2021 for the premontane region in the eastern part of the study area. We have added this information in the text.

Lines 124-126:
"Lantmäteriet (2021a) provides natural colour (RGB 4, 3, 2) and colour-infrared (RGB 5, 4, 3) orthophotos with a resolution of 0.5 m acquired in 2018 and 2021 for roughly the western montane region and eastern premontane region, respectively (Fig. 1c)."

New reference:
"Lantmäteriet: Product description: Orthophoto (Document version 2.6), https://www.lantmateriet.se/globalassets/geodata/geodataprodukter/ flyg--och-satellitbilder/e_pb_ortofoto.pdf, 2021a."

- *Lines 139–145: "The application of an inversion model is required for extracting ice-sheet properties from mapped glacial geomorphology, such as its thermal regime, subglacial hydrology, or the presence of ice streams. Here, the conceptual framework of Kleman et al. (2006) is applied to deduce ice sheet evolution through time. The inversion model is composed of a set of assumptions (Kleman et al., 2006, p. 196), a classification system for glacial landform assemblages, and a procedure for managing the landform data and incorporating absolute chronological data. The model thus explains how individual landforms are interpreted in terms of ice sheet properties, which results in ice sheet-wide glaciologically-consistent patterns by aggregation of the individual landforms into swarms."*

The mapping approach is also described relatively clearly, however you may wish to consider adding more detail here on the inversion model (Kleman et al., 2006) to ensure sufficient methodological detail is provided without the requirement to read that chapter.

Given the aim of our study, a paragraph was added where the approach by Kleman et al. (2006) is summarized for identifying deglacial swarms. This should ensure sufficient methodological detail to understand the approach to our reconstruction as outlined in Figure 9. Additionally, in response to a comment by reviewer 2, an extra column was incorporated in Table 1 summarizing the mapping approach. Table 1 already described for each landform its paleoglaciological significance, but now additionally describes how the landform was mapped.

Lines 158-165:

"Wet-bed deglaciation swarms include eskers with aligned lineations. These fields of lineations and eskers are formed time-transgressively, parallel to ice flow, and perpendicular to the ice margin. Dry-bed deglaciation swarms typically lack subglacial landforms, due to an absence of sliding when the ice sheet is frozen to its substrate, but include meltwater channels, ice-dammed lake shorelines, and perched deltas. Such meltwater traces are imprinted on a relict surface, which can be non-glacial or glacial, thus demonstrating the subglacial preservation of landforms and landscapes. Ribbed moraine forms when subglacial conditions change from dry to wet-bed (Hättestrand, 1999), with its individual ridges oriented perpendicular to ice flow direction. A set of these landforms representing coherent ice flow directions and ice margins can then be outlined to realistically visualize retreat patterns."

- *Figures 3-5*

I would suggest including a few extra words in the captions for figures 3, 4 and 5 when revising this manuscript so the first sentence of the caption states what type of glacial landforms are shown in each selection of locations. For example, for figure 4 you could state "Examples of landforms associated with ice-marginal lakes in the study region".

This is a good suggestion, we have incorporated this in our Figures 3, 4 and 5. Additionally, we now formatted every first sentence of all our figure captions in bold for enhanced clarity.

Figure 3, after line 210:
"Examples of subglacial, ice-marginal, and proglacial landforms in the study region."

Figure 4, after line 273:
"Examples of landforms associated with ice-dammed lakes in the study region."

Figure 5, after line 287:
"Examples of outlet channels associated with ice-dammed lakes in the study region."

**RC1 – Colby Smith**

**General comments**
The manuscript is well organized around the geomorphic map, and draws conclusions related to the timing and dynamics of both the ice sheet retreat and the associated glacially induced faulting in northwestern Sweden. I recommend that the manuscript be published after minor revisions.

I have made some minor comments related to language, grammar, or general readability. These are included in the attached pdf of the manuscript.

All linguistic changes were adopted and clearly improved the general readability of the manuscript. We would like to express our thanks for the reviewers' thorough check of the language.

Here, I will focus on the single larger issue that I believe needs to be addressed prior to publication. The authors make a case that different segments of the Pärvie fault ruptured at

different times during or shortly after the late Weichselian deglaciation. Along Torneträsk, the authors present a strong case for fault rupture shortly after deglaciation based on the fact that older shorelines are displaced by the fault but not younger shorelines. Farther south, the other examples of where faulting either precedes or follows deglaciation are not as well presented. I do not question your conclusions, but that is because I have spent a lot of time looking at these faults in LiDAR. Other readers may need further convincing.

The crosscutting relationships are not obvious in figure 10 and they are not adequately described in the text. Thus, I suggest revising figure 10 to be more like figure 3 and clearly show the crosscutting relationships between the fault and glacial landforms, and the fault and the shorelines. Additionally, include text that explains the crosscutting relationships and the conclusions drawn from them.
This is a nice contribution to the deglacial history of northern Sweden, and I look forward to seeing it published.

First, we want to thank the reviewer for his kind words and encouragement. We are pleased that our work is considered a contribution to the deglacial history of northern Sweden. Second, we thank the reviewer for pointing out the issue regarding the crosscutting relationships and for giving us the opportunity to revise Figure 10. We provide a detailed response and revisions further down in the document, where we present an improved Figure 10 and an additional panel to Figure 6.

**Specific comments (as annotated in PDF)**

- *Lines 74-76: "Finally, a refined history of ice-marginal retreat potentially enables future investigations of the interaction between the retreating ice sheet and the re-activation of faults through glacial isostatic adjustment (Fig. 1b)."*

The primary stress related to these faults is tectonic (ie related to the spreading of the Atlantic). Glacial isostatic adjustment is more the trigger than the cause. See: Stewart, I.S., Sauber, J. & Rose, J., 2000: Glacio-seismotectonics: ice sheets, crustal deformation and seismicity, Quaternary Science Reviews 19, 1367–1389.

The sentence has been adjusted to emphasize how the total stress field is comprised of both the presence of a regional tectonic stress and the superimposed glacially-induced stress.

Lines 74-77:
"Finally, a refined history of ice-marginal retreat potentially enables future investigations of the interaction between the changing configuration of the retreating ice sheet and its marginal positions and a re-activation of faults through the overprinting of the prevailing regional tectonic stress with glacially-induced stress".

- *Line 80: "The Torneträsk Basin cuts through the Scandinavian mountain range, also known as the Scandes, and across the border to Norway forms the headwaters of Rombaksfjorden (Rombaken; Fig. 2)."*

There is no subject in this clause. It is incomplete. What forms the headwaters? It's also unclear because it sounds like the Torneträsk basin is the headwaters of the fjord. Torneträsk drains to the east, the fjord is over a pass to the west.

The sentence is altered to emphasize that Torneträsk basin is draining to the east. The confusion probably stemmed from Fig. 2 showing that during deglaciation the Fennoscandian Ice Sheet drained westwards across the Torneträsk basin and across the water divide, providing meltwater to Rombaksfjorden.

Lines 80-82:
"The Torneträsk Basin cuts through the Scandinavian mountain range, also known as the Scandes, and drains to the east. Across the border to Norway, over a pass to the west, are the headwaters of Rombaksfjorden (Rombaken; Fig. 2)."

- *Line 119: "Additional azimuths of 90°, ° and 180°"...*

Should there be another direction here?

Thank you for finding this mistake! The degree symbol is removed, there was no other azimuth used than the two mentioned.

Line 122:
"Additional azimuths of 90° and 180°"

- *Line 259: "... there are no other geomorphological cross-cutting relationships that show the exact offset as well as the raised shorelines."*

This is true, but the crosscutting relationship between fault and shoreline is not visible in the LiDAR imagery.

We suspect the confusion stems from the use of the phrase "as well as", which was meant as "and also", not as "as good as". The crosscutting relationship between the fault and shorelines is only evident from regional analyses using the graph that plots the shoreline elevations along a reference plane (old and updated Figure 6, below), not from the LiDAR imagery itself due to the lack of continuity of the shorelines at the location of the fault scarp. This study is basically outlining a new technique to identify fault ruptures. The sentence is changed to emphasize that the crosscutting is not visible in the LiDAR imagery.

Lines 282-283:
"... there are no geomorphological cross-cutting relationships visible in the LiDAR imagery that show the offset of raised shorelines at the exact location of the fault scarp (Fig. 6b)."

[Figure]

**Figure 6. Lake stages of ice-dammed lake Torneträsk.** (a) Individual lake levels were identified from the elevations of raised shorelines, perched deltas, and outlet channels. At the Abscissa value of zero, the ordinate value is 342 m a.s.l., the current elevation of the surface of Torneträsk. The approximate location where the Pärvie Fault crosscuts the Torneträsk Basin is indicated by the red bar. The distance is calculated along an axis perpendicular to the isobases

of postglacial rebound of the shorelines (see Fig. 1c). The corresponding elevation ranges are summarised in Table 2. (b) Elevations of raised shorelines of ice-dammed lake Torneträsk on either side of the Pärvie Fault where it crosscuts the northern shore of Torneträsk (see red bar in (a)), illustrating elevation jumps of around 8 m for the higher raised shorelines (T3-T6), while the lowest raised shoreline (T7) crosses the fault at 365–366 m a.s.l. The background is a shaded relief based on the DEM provided by ©Lantmäteriet. See location in Fig. 1c.

- *Lines 260-261: "Remarkably, ice-dammed lakes T1 and T2 do not appear to have been offset by the Pärvie Fault, although this has perhaps remained unrecorded due to a lack of shoreline segments for these lake stages."*

I would revise this to indicate that these older shorelines do not record the fault displacement because of the lower resolution of the data.  If there were continuous shorelines they would be crosscut by the fault.

We agree that the shorelines would be crosscut by the fault if they were continuous. The sentence is adjusted to emphasize this argument.

Line 284:
"Ice-dammed lakes T1 and T2 do not record the offset by the Pärvie Fault due to the lower resolution of the data (Fig. 6)."

- *Line 283-284: "Mapping of surficial geology by the Geological Survey of Sweden (SGU, 2024a) shows that"* …

This is technically cited correctly if it came from the database this year. It is, however, significant that the mapping was carried out on aerial photos not lidar (in the 1990s I think).

First, while working on this comment, we concluded that the in-text citation referred mistakenly to the wrong SGU dataset on the highest shoreline. Second, the original URL of the correct SGU dataset linked to a pdf with a product description of a surficial geology dataset at another scale. Both these errors are corrected in the text and in the reference list.

Updated reference:
"SGU: Product description: Surficial geology 1:250 000, northernmost Sweden (Swedish), https://resource.sgu.se/dokument/produkter/ jordarter-250000-nordligaste-sverige-beskrivning.pdf, 2024a."

We agree that the reference could be misleading to infer that the work in it was performed recently. However, it is correct to cite a product description from SGU which was last updated in 2024. Its maps are based on compilation and digitization of older surveys (the document does not clarify from which years), where the mapping was indeed mainly based on aerial photo interpretation together with field observations along the sparse road network. Elevation models were not used. The compilation was finished in 2011. In a separate guide by SGU to the surficial geology maps and databases of Sweden, it appears they started to use aerial imagery for mapping in the 1980s. In the digital map viewer, it shows that most of the mapping was finished in early 2000s.

Lines 306-309:
"Mapping of surficial geology by the Geological Survey of Sweden (SGU, 2024a), carried out from the 1980s to early 2000s based on aerial imagery supported by sparse field observations, shows that vast amounts of glacio-fluvial sediments which form deltaic deposits are found 165 km downstream of the initial drainage location(s) along both Tornedalen and Kalixdalen (Fig. 8d)."

- *Lines 379-382: "These are favorable comparisons because GLOFs from Akkajaure and Sitojaure cut the Ancylus Lake highest coastline and are therefore in timing close to the youngest GLOF of Torneträsk (and so are their shoreline gradients) and lake evolution in central Jämtland spans 10.5–9.2 cal ka BP (Regnéll et al., 2023), which brackets ice retreat from the Torneträsk Basin, rendering it reasonable that the gradients of IDLT fall within the range of values from central Jämtland."*

This is too much information for one sentence. Break this up into smaller sentences and move the parenthetical text into the body of a sentence.

We appreciate the opportunity to revise this section. The study from Regnéll et al. (2019) did not trace the GLOFs from Akkajaure and Sitojaure (the two lakes we cited), but Pieskehaure and Mavasjaure, which are approx. 90 km farther south. These two ice-dammed lakes were, however, not reconstructed, so there are no shoreline gradients available to compare to. We therefore removed this statement from our argumentation. Breaking up the sentence as the reviewer suggested improved the readability as well. The entire paragraph was moved to the Chronology section in response to a comment by reviewer 2.

Lines 493-497:
"These are favorable comparisons because ice-dammed lakes Akkajaure and Sitojaure also formed in response to the final deglaciation of the Fennoscandian Ice Sheet. Their timing is closest to the youngest GLOF of Torneträsk, and so are their shoreline gradients. Additionally, lake evolution in central Jämtland spans 10.5–9.2 cal ka BP (Regnéll et al., 2023), which brackets ice retreat from the Torneträsk Basin, rendering it reasonable that the gradients of IDLT fall within the range of values from central Jämtland."

- *Line 386: "The reconstruction of the ice-dammed lakes in the Torneträsk Basin using LiDAR resulted in eight stages"…*

Add a sentence at the beginning of this paragraph that allows the reader to know where you headed. "Previously published work on ice-dammed lakes in the Torneträsk basin are broadly consistent with the results of this study but lack the detail allowed by use of the DEM."

Thank you for the helpful suggestion. We have slightly altered the suggested topic sentence and added it to the beginning of a new paragraph that was written in response to reviewer 2, where we compare our general mapping, in addition to the ice-dammed lakes, to other glacial geomorphological studies. These paragraphs are now in their own separate section in the Discussion.

Lines 559-560:
"Previously published work on glacial geomorphology in the Torneträsk Basin are broadly consistent with the results of this study but lack the detail allowed by the use of the LiDAR-based DEM".

- *Lines 432-435: "The amplifying effect that ice-marginal lakes have on retreat rates of lacustrine-terminating ice sheets (e.g., Stokes and Clark, 2004; Utting and Atkinson, 2019), explains that the FIS experienced higher ice losses due to the ice-dammed lakes in the Torneträsk region, and helps explaining the pivoting motion of ice retreat in this region."*

Flip this sentence around. "Higher ice losses due to the ice-dammed lakes in the Torneträsk region can be explained by the amplifying effect...... Thus, the presence of the ice-dammed lakes led in part to the pivoting…"

This suggestion indeed helps improve readability.

Lines 420-424:
"Higher ice losses of the FIS due to the ice-dammed lakes in the Torneträsk region are consistent with the amplifying effect that ice-marginal lakes have on retreat rates (e.g., Stokes and Clark, 2004; Utting and Atkinson, 2019). Thus, the presence of the ice-dammed lakes led in part to the pivoting motion of ice retreat in this region."

- *Line 437-438: "The ice-marginal positions that dammed the successive ice-dammed lake stages of Torneträsk fall approximately in-between their 10.1 and >9.9 cal ka BP isochrons (Fig. 1b)"…*

How many dates actually constrain these isochrons?

Few, if any; there is a real dearth of data in this region. See Hughes et al. (2016) and Stroeven et al. (2016) for data compilations and resulting deglaciation histories.

- *Lines 460-461: "In fact, the fault may have ruptured subglacially as well while as much as 95 km of the fault trace was ice covered at that point."*

If the fault ruptured along its entire length at this time, then as much as 95 km.......

Thank you for pointing this out, because it is indeed questionable whether the fault ruptured along its entire length.

 Lines 435-436:
"In fact, if the fault ruptured along its entire length at this time, then as much as 95 km of the fault trace were ice covered at that point."

- *Line 466: … "and pre-date deglaciation south of this ice sheet margin (Fig. 10)."*

As originally suggested by Lundqvist and Lagerbäck (1976).

Citing this paper highlights how the findings of our study are in contrast to the original school of thought.

Lines 438-441:
"If all of the Pärvie Fault ruptured at once, such as is typically considered when calculating the amount of energy released, cross-cutting should post-date deglaciation north of the inferred ice sheet margin at the time of rupture (between T6 and T7, Figs. 9e, f) and pre-date deglaciation south of this ice sheet margin, as originally suggested by Lundqvist and Lagerbäck (1976)."

- *Figure 10*

I can't see these crosscutting relationships in the images provided. I suggest a revised figure similar to Fig. 3 with larger, clearer lidar images (perhaps without slope?). Additionally, to demonstrate the uncertainty involved here, include a lidar image of your crosscut shorelines. You present a convincing case that the shorelines are cut by the fault, but I can't see it in the imagery alone. This should be presented and discussed in the text as well.

We thank the reviewer for giving us the opportunity to revise Figure 10 to the standard of Figure 3. We have followed his suggestion, almost entirely. Below the former Figure 10 map, we now indeed present four panels with significantly improved LiDAR imagery showing the cross-cutting relationships referenced in the manuscript. The four panels show one example of the Pärvie fault postdating (cutting) the glacial landforms or postglacial fluvial landforms, one instance where an esker drapes the fault trace, thus indicating faulting before esker formation, and two panels where we tentatively conclude that fluvial terraces have been offset multiple times. What is not included is a panel that shows the Pärvie fault cutting the shorelines and off-setting them on either side of the fault trace. If this evidence existed, we suspect this information might have been presented earlier-on. Rather, it is the painstaking reconstruction of the shoreline fault traces along the full length of the Torneträsk basin and across the full range of elevations that shows this jump in elevation of the oldest shorelines at the location of the Pärvie Fault. In response to an earlier comment, we added an additional map that is presented together with former Fig. 6, which illustrates the shoreline traces that are visible in the immediate surroundings of the Pärvie Fault trace (<1 km) and the elevation jumps (~8m) that can be gleaned from these.

We believe that improved Figures 6 and 10 and their informative captions cover the information sought by the reviewer, and we abstain from further inclusion of a discussion paragraph as the imagery speaks the language and because this information aligns with excellent papers written by the reviewer himself (Smith et al., 2018, 2021), to which we direct the readers if they are interested in the Pärvie Fault.

Old Figure 10:

[Figure]

New Figure 10:

[Figure]

**Figure 10. Inferred cross-cutting relationships between glacial geomorphology and the Pärvie Fault.** (a) Compilation of sites within the study area of complex relationships between expressions of the Pärvie Fault and landforms including faulting (b) pre-dating deglaciation, where an esker drapes a fault scarp, (c) post-dating deglaciation, where glaciofluvial landforms are cut by a fault scarp, and (d-e) occurring, tentatively, multiple times, where fluvial terraces are offset by multiple ruptures. Panel (d) portrays the same location as in Smith et al. (2021, Fig 12.4). The background is a shaded relief based on the DEM provided by ©Lantmäteriet.

- *Line 480: "Currently, such an approach appears unrealistic given the mounting evidence for different types of cross-cutting relationships (Fig. 10), reinforcing observations by Lundqvist and Lagerbäck (1976) that the Pärvie Fault ruptured both prior to, and after, deglaciation."*

This was published before people knew that there were multiple generations of glacial landforms in N. Sweden. Thus, their conclusion was that a single rupture occurred partially under the ice.

Thank you for this reminder. To avoid confusion, this citation is removed together with the statement about the rupture being both prior and after deglaciation. Instead, we added a sentence emphasizing how the cross-cutting evidence reinforces alternative interpretations of multiple ruptures.

Lines 453-455:
"Currently, such an approach appears unrealistic given the mounting evidence for different types of cross-cutting relationships (Fig. 10), reinforcing observations that the Pärvie Fault ruptured multiple times."

- *Line 535-536: "Cross-cutting relationships between glacial landforms and fault scarps indicate that the Pärvie Fault ruptured multiple times during the last deglaciation and, indeed, before the last deglaciation."*

What do you mean by this? Before the last glaciation was complete?
You do not discuss evidence of pre-late Weichselian faulting.

Indeed, we have no evidence for pre-late Weichselian faulting. We only have evidence of faulting during deglaciation, where the rupture happened shortly after ice retreat. We remove the latter part of the sentence that insinuated rupture before the last deglaciation.

Lines 603-605:
"Cross-cutting relationships between glacial landforms and fault scarps indicate that the Pärvie Fault ruptured multiple times during the last deglaciation, and close to the retreating ice margin."

**RC2 – Benjamin Boyes**

**General remarks**

This article presents a new framework for understanding late glacial landscape evolution in northern Sweden. The study uses original geomorphological data and previously published chronometric data to reconstruct Fennoscandian Ice Sheet retreat patterns, ice dammed lake development, and the evolution of post-glacial faults. This publication is suitable for publication in *The Cryosphere* after minor revisions, and I look forward to seeing it published.

We thank the reviewer for his encouraging feedback and for considering our work a valuable contribution to the understanding of late glacial landscape evolution in northern Sweden. We are pleased that the reviewer finds our study fit for publication.

*Academic rigour and accuracy*
The study's methodology is comprehensive, and the results are clearly laid out. However, it would be useful if the following points are clarified:

- The mapping methods could be more clearly laid out. The manuscript suggests you mapped a wide array of features, but details (e.g. how polylines are drawn to map landforms) are only provided for ice dammed lakes (and associated features).

The mapping approach for the other landforms is described in an extra column in Table 1, while the more comprehensive description for the ice-dammed lake traces is kept in the text as it was. We added a sentence referring to Table 1 for the details on the mapping approach.

Lines 166-167:
"The mapping approach, that is, how the landforms are delimited in GIS software, is briefly described for all landforms in Table 1. Given the focus on ice-dammed lake traces, the mapping approach of raised shorelines and perched deltas, and the methodology to identify ice-dammed lake stages, are described in more detail below."

- The fault lines and rock slope failure deposits are not presented in the results. These data are from previous work (as suggested by Figure 1b) and the source of these data need to be more obviously discussed in the text. If you checked these against the LiDAR data, this needs to be discussed.

Thank you for this suggestion. We have added a paragraph in the Methods/Datasets section to discuss the data sources in more detail.

Lines 130-137:
"Vector data sets of previously published studies were used for different purposes. The international database of Munier et al. (2020) contains glacially-induced faults in northern Fennoscandia (Fig. 1b), of which many were previously proposed and recently confirmed based on the recent LiDAR data. The faults in the database were cross-referenced with the LiDAR-based DEM, but no effort was made to identify new faults. The dataset was used to identify cross-cutting relationships between glacial landforms and fault scarps. The deglaciation isochrons reconstructed by Stroeven et al. (2016) were used to evaluate the implications of the direction of mapped landforms and to constrain the chronology (Fig. 1b). Cosmogenic nuclide [10]Be exposure ages of two rock slope failure (RSF) deposits were taken from Stroeven et al. (2002, 2016). The RSF extents were cross-referenced against the LiDAR-based DEM."

- The mapping is good and has added considerable detail to the region, and I like how clear the supplementary map is. However, from personal experience mapping landforms in this region from similar LiDAR data and in the field, I think some features have not been mapped. This is entirely subjective, but it would be good to know why you chose to map certain features and if you chose to omit any?

We are not sure whether this comment is referring to entire feature classes or to individual features: in our answer we presume that latter. Although there is of course the aim to identify all features, there were certainly features where the actual landform type remained ambiguous. In this respect the map is conservative: we only mapped features of which the genetic interpretation could be confidently determined. Given that dataset, we were able to draw robust conclusions, which are insensitive to the total number of mapped features.

- The relative timing (e.g. last glacial vs previous glacial) of some of the features needs clarification. Why have you decided which landforms are pre-last glacial, and maybe show these features on a map of their own? You mention that this is a thing, but don't provide any evidence of pre-last glacial landforms.

We appreciate the reviewer's comment regarding the relative timing of our mapped features. We understand the importance of distinguishing between last glacial and pre-last glacial landforms. However, we do not believe that we explicitly stated it as a significant issue ("a thing") in our manuscript. Our intention was to provide a general context, as we did in the following sections:

- **Study area section (former lines 95-103, now 97-105 )**: We describe that the area is known for its palimpsest landforms.
- **Discussion/Ice-marginal positions section (former lines 411-413, now 399-401)**: We reiterate that it is known that landforms are not exclusively from the last deglaciation.
- **Discussion/Glacially-induced faulting section (former lines 470-475, now 445-450)**: We mention that traces of older glaciations can complicate reconstructions, such as those of relative timing between fault rupture and glacial landforms. We mention that glacial landforms cut by the Pärvie Fault within the study area align with reconstructed deglaciation directions.

We are uncertain about the improvements the reviewer is suggesting. After careful consideration, we believe that the current presentation aligns best with the overall objectives of

our manuscript. Therefore, we will retain the original content on this topic. We thank the reviewer for making us re-think this topic.

- As I understand it, mkm$^{-1}$ is a unit referring to slope? A short sentence clarifying what this means would be helpful.

Thank you for pointing this out: it is indeed a unit referring to the gradient of the shorelines. A sentence is added to the Methods chapter to explain the unit.

Lines 177-178:
"The tilting of the shorelines is described as a gradient in m km$^{-1}$ where the elevation difference (in meters) is given over the distance (in kilometers) in the direction of the reference plane."

- Some discussion on how your geomorphological mapping compares with existing geomorphological maps could be interesting. You do this comprehensively for the lakes, but not the other landforms.

This is a good idea. It requires two steps. First, we need to explain what data sources we have for comparison and how they were digitized, and then we discuss how our mapping compares to the previously published maps in a new section in the Discussion chapter.

A statement on the inclusion of printed maps for cross-referencing is included in the Methods chapter in lines 137-139:
"The printed geomorphological maps by Melander (1977a, b) and Hättestrand (1998) were digitized and georeferenced in GIS software using locations on the map with known coordinates for cross-referencing purposes."

We have added another paragraph to our Discussion where we present a global comparison between our mapping and previous maps. We moved this paragraph and our previously written comparison for the ice-dammed lakes (former lines 386-401, now 575-591) to a new section at the end of our Discussion.

Lines 561-574:
"The most detailed geomorphological maps of the Torneträsk region were produced by Melander (1977a, 1977b). His landform interpretation is based on aerial photographs and extensive field verification, and resulted in a comprehensive geomorphological map presented at 1:250,000. Our mapping (Fig. S1) is consistent with his mapping but adds considerable detail in terms of the number of raised shorelines (resulting in more ice-dammed lake stages), and the number of channels in flights of lateral meltwater channels. Additionally, whereas we map different types of meltwater channels, Melander (1977a, 1977b) only categorizes glaciofluvial channels by size. For example, some large glaciofluvial channels correspond to outlet channels of ice-dammed lakes in this study. A critical difference between our maps is the number of lineations; our mapping includes significantly more lineations in both the premontane and the montane regions. The last glacial geomorphological map covering the Torneträsk region was produced from aerial photographs by Hättestrand (1998) at 1:1,250,000. Unlike Melander (1977a, b), this map includes large and small scale lineations, ribbed moraine, DeGeer moraine,

and Veiki moraine. Our landform distributions of those features are consistent with the Hättestrand (1998) map but provide more detail, as individual lineations are outlined rather than a representative for a larger area. Thus, our mapping based on high-resolution LiDAR data, as expected, adds more detail in terms of landform count but is consistent with previously-mapped landform distributions. The critical implication of added detail in our mapping resides in a more detailed reconstruction of the ice-dammed lakes, but does not alter general inferences on ice retreat from ice flow directional indicators."

During revision of this section, we added a few lines to highlight some of our new findings, particularly the GLOFs and the Pärvie Fault cutting most, but not all of the Torneträsk raised shorelines.

Lines 590-594:
"The geomorphic traces of the GLOFs of IDLT stages T6–T8, which terminated in Ancylus Lake, were not described before. Although the Pärvie Fault has been the subject of much investigation (Lundqvist and Lagerbäck, 1976), the cross-cutting relationship between the Pärvie Fault and the oldest raised shorelines in the Torneträsk Basin has only become evident thanks to a regional analysis of shoreline gradients facilitated by recently released LiDAR data (Fig. 6)."

- Table 1: This is a nice comprehensive table – I particularly like the "possible identification error" column. It would be helpful to have in this table a column that explains the mapping approach, as an example figure in e.g. Boyes et al., 2021 (https://doi.org/10.1080/17445647.2021.1970036) and/or as text.

This suggestion improved the description of our mapping approach. We have inserted the mapping approach as the next-to-last column in Table 1. We also used this suggestion as an opportunity to highlight references to figure examples in the Morphology column by formatting them in a bold font.

[revised manuscript text omitted]

- Figure 1: In panel b, consider thinning out the isochrons or making the panel bigger. At present, it's a little difficult to see all of the components in the figure.

We enlarged panels a and b and thinned out the isochrons and country borders in panel b, which made it easier to appreciate all components in the figure.

Old Figure 1:

[Figure]

New Figure 1, after line 87:

[Figure]

- Figure 9: If you are unsure whether the ice sheet also retreated into the Kebnekaise/Sarek Mountains, consider leaving a ? symbol over these locations in your retreat pattern to acknowledge this.

We have added another question mark in panel g over the mountains of Kebnekaise, which better visualizes that the retreat at this location remains unconstrained. The Sarek Mountains are well outside the mapping boundaries. We also enlarged the arrows representing outlet channels to increase their visibility.

New Figure 9, after line 401:

[Figure]

- Figure 10: The cross-cutting is really difficult to see. Make the panels bigger with nice and clear LiDAR hillshade images.

We thank the reviewer for his comment, which has also been raised by reviewer 1. We present an improved Figure 10 below.

Old Figure 10:

[Figure]

New Figure 10, after line 469:

[Figure]

**Figure 10. Inferred cross-cutting relationships between geomorphology and the Pärvie Fault.** (a) Compilation of sites within the study area of complex relationships between expressions of the Pärvie Fault and landforms including faulting (b) pre-dating deglaciation, where an esker drapes a fault scarp, (c) post-dating deglaciation, where glaciofluvial landforms are cut by a fault scarp, and (d-e) occurring, tentatively, multiple times, where fluvial terraces are offset by multiple ruptures. Panel (d) portrays the same location as in Smith et al. (2021, Fig 12.4). The background is a shaded relief based on the DEM provided by ©Lantmäteriet.

- *Line 119: ... "as these are considered the optimal values for the visualization of hillshade relief models for the purpose of glacial geomorphological mapping (Chandler et al., 2018)."*

Chandler et al., 2018 don't suggest these values for hillshade images, other authors do (specifically Chandler cite Smith and Clark, 2005 and Hughes et al., 2010). Change (or add) the citation to other sources.

Thank you for spotting that, the older citations are adopted.

Lines 119-122:
"The DEM was processed in ArcGIS Pro 2.9.3. to create a hillshade relief model using an illumination angle with an altitude of 30° and azimuths of 45° and 315°, as these are considered the optimal values for the visualization of hillshade relief models for the purpose of glacial geomorphological mapping (Smith and Clark, 2005; Hughes et al., 2010)."

- *Lines 119-121: "Additional azimuths of 90°, ° and 180°, perpendicular and parallel to the dominant lineation orientation, respectively, were applied to reduce the 'azimuth bias' (Smith and Clark, 2005; Chandler et al., 2018)."*

 Either remove the statement or define which azimuths were used.

The degree symbol has been removed, there was no other azimuth used than the two already mentioned. Reviewer 1 had the same comment, thank you for spotting this.

Line 122:
"Additional azimuths of 90° and 180°"

- *Lines 164-167: "The glacial geomorphology of the Torneträsk Basin is presented in Fig. S1. The total comes to 6633 mapped features, of which there are 2796 lineations, 678 eskers, 39 ribbed moraine, 1262 meltwater channels, 155 marginal moraines, 510 undifferentiated moraines, 894 raised shorelines, 206 perched deltas, 25 outlet channels, and 38 veiki moraines. Note that the count includes all segments of a landform, so it represents a feature count instead of a landform count."*

Here you provide numbers of how many features you have mapped. However, because you have not detailed the mapping approach for each landform type, it is not clear whether the quoted 6,633 mapped features are individual features or groups of features. For example, you say you have mapped 38 veiki moraines – is that 38 areas of veiki moraine, or 38 individual veiki moraine plateau?

The mapping approach is now described in Table 1, which clarifies that ribbed moraine and veiki moraine are mapped as areas, rather than individual ridges or plateaus. The text is also changed to emphasize that the feature count refers to the number of ribbed moraine and veiki moraine areas.

Lines 187-189:
"The total comes to 6633 mapped features, of which there are 2796 lineations, 678 esker segments, 39 areas of ribbed moraine, 1262 meltwater channels, 155 marginal moraines, 510 undifferentiated moraines, 894 raised shorelines, 206 perched deltas, 25 outlet channels, and 38 areas of veiki moraine."

Later on in the results section, you go on to say landforms are found "relatively often" in x locations. It would be better to put a number (i.e. %) on this.

We now present percentages of landforms in relation to their proportion in the montane and the premontane regions, in their respective Results sections.

Line 193:
 "Lineations occur across the area but are most common in the premontane region (60%, Fig. S1)."

Line 202:
"Eskers occur across the area, but are most frequent in the montane region (63%)."

Lines 210-211:
"Subglacial meltwater channels are prevalent in the entire study area, although most of them occur in the premontane region (72%, Fig. S1)."

Lines 216-217:
"Ribbed moraine occurs predominantly in the premontane region (56%) and in the montane region on uplands north of Torneträsk and in between Rautasjaure and Torneträsk (Fig. S1)."

Line 227:
"Whereas subglacial channels are abundant in the premontane region, lateral meltwater channels are relatively rare (21%)."

Line 246:
"Moraines are virtually lacking in the premontane region (4%, Fig. S1)" …

- *Lines 257-259: "However, whereas there is abundant information on fault displacement of glacial geomorphology, indicating that the Pärvie Fault ruptured after landform formation (Figs. 3b and 3c), there are no other geomorphological cross-cutting relationships that show the exact offset as well as the raised shorelines."*

Please point to this cross-cutting relationship on the figure.

Thank you for asking clarification on this, reviewer 1 commented on the same lines. The crosscutting relationship between the fault and shorelines is only evident from regional analyses using the graph that plots the shoreline elevations along a reference plane, not from

the LiDAR imagery itself due to the lack of continuity of the shorelines at the location of the fault scarp. This study is basically outlining a new technique to identify fault ruptures. Additionally, a new Figure 6 is presented to show the raised shorelines at the location where the fault crosscuts the basin.

Lines 282-283:
"there are no geomorphological cross-cutting relationships visible in the LiDAR imagery that show the offset of raised shorelines at the exact location of the fault scarp."

Old Figure 6:

[Figure]

New Figure 6, before line 288:

[Figure]

[Figure]

**Figure 6. Lake stages of ice-dammed lake Torneträsk.** (a) Individual lake levels were identified from the elevations of raised shorelines, perched deltas, and outlet channels. At the Abscissa value of zero, the ordinate value is 342 m a.s.l., the current elevation of the surface of Torneträsk. The approximate location where the Pärvie Fault crosscuts the Torneträsk Basin is indicated by the red bar. The distance is calculated along an axis perpendicular to the isobases of postglacial rebound of the shorelines (see Fig. 1c). The corresponding elevation ranges are summarised in Table 2. (b) Elevations of raised shorelines of ice-dammed lake Torneträsk on either side of the Pärvie Fault where it crosscuts the northern shore of Torneträsk (see red bar in (a)), illustrating elevation jumps of around 8 m for the higher raised shorelines (T3-T6), while the lowest raised shoreline (T7) crosses the fault at 365–366 m a.s.l. The background is a shaded relief based on the DEM provided by ©Lantmäteriet.

- *Lines 419-423: "Hence, a strong control of topography on ice retreat patterns and rates is evident, as other studies have demonstrated for the FIS (Stroeven et al., 2016; Szuman et al., 2024), the British-Irish Ice Sheet (Greenwood et al., 2007; Hughes et al., 2014), and the Cordilleran Ice Sheet (Kleman et al., 2010; Dulfer et al., 2022)."*

Topographic controls on ice sheet geometry during retreat of a thinning ice sheet have also been highlighted in northwest Russia (https://doi.org/10.1002/jqs.1130; https://doi.org/10.1016/j.quascirev.2022.107872; https://doi.org/10.1111/bor.12653).

Thank you for these suggestions. We added another citation to represent the deglaciation of the northwestern sector of the Fennoscandian Ice Sheet.

Lines 407-409:
"Hence, a strong control of topography on ice retreat patterns and rates is evident, as other studies have demonstrated for the FIS (Stroeven et al., 2016; Boyes et al., 2023; Szuman et al., 2024),.."

- *Lines 436-439: "In the reconstruction of Stroeven et al. (2016), the retreating ice margin swept across the study area in a time span of 500yr (Fig. 1b). The ice-marginal positions that dammed the successive ice-dammed lake stages of Torneträsk fall approximately in-between their 10.1 and >9.9 cal ka BP isochrons (Fig. 1b), which would suggest the ice-dammed lake system of Torneträsk existed for a total duration of <200 yr."*

You briefly mention timing of lakes here and have more detail on faulting chronology in Section 5.4. Could you have a single chronology section that deals with the chronologies of each component (ice sheet retreat, ice dammed lake formation/drainage, and faulting) as they are interlinked.

We thank the reviewer for this excellent suggestion and agree that the chronologies of each component in this reconstruction are interlinked and conclude that a single chronology section improves the structure of the paper. We have consolidated the chronologies of each component into a single section. We use the opportunity of revision to introduce cosmogenic in situ $^{14}$C dating, which was recently demonstrated to be a promising technique for constraining deglaciation ages in Sweden by Goodfellow et al. (2024). We suggest using cosmogenic in situ $^{14}$C dating of the exposed bedrock of the outlet channel of IDLT6 (and spillway of IDLT7) to constrain the timing of the Pärvie Fault rupture.

Lines 528-532:
"Finally, because the outlet of IDLT stage T6 (western channel in Fig. 5c) also was the spillway channel of IDLT stage T7, bedrock in this channel would be first fully exposed to cosmic rays after failure of the IDLT stage T7 ice dam only a decade or decades after the rupture of the Pärvie Fault. Cosmogenic in situ [14]C dating of outlet channel bedrock is likely the most direct methodology to determine the age of faulting (within uncertainty). The promise of this technique to deliver accurate deglaciation ages in Sweden was demonstrated by Goodfellow et al. (2024)."

It would be better to use the point chronometric data presented by Stroeven et al., 2016 and in the DATED-1 database rather than comparing to the isochrons as this may provide more relevant information for your reconstruction.

The point chronometric data of Stroeven et al. (2016) and the DATED-1 database present challenges for comparison with our mapping due to limited constraints over a large area. Stroeven et al. (2016) lack landform types, making it difficult to draw any conclusions about ages without going into the geomorphological context of every sample individually. Many samples are from bedrock, representing cumulative exposure from previous ice-free periods. The DATED-1 database includes only seven individual ages within our study area, of which four are deglacial. Three of these deglacial ages are radiocarbon dates from the same moraine-dammed lake, of which the location is stored incorrect in the database.

Given the scarcity of data in our region, we believe that a comparison with point chronometric data would not significantly enhance our reconstruction. We therefore refrain from implementing the suggested changes.

- *Line 515: "There are two large rock slope failure (RSF) deposits in the study area that were potentially triggered by ruptures along the Pärvie Fault."*

You've suggested that the rock slope failure deposits are a result of post-glacial earthquakes. Such landslides can also be triggered by glacial de-buttressing during glacier retreat. You should include some discussion on this point, and if you still consider these landslides to be earthquake induced, then you need to clearly provide evidence for this.

We agree that this statement needs to be discussed, and we added a paragraph in our Discussion, arguing that we cannot conclude whether the rock slope failure deposits are earthquake-induced or the product from other processes. The comment also inspired to look for more geomorphological evidence regarding subglacial fault rupture.

Lines 533-557:
"The absence of a larger group of landslides in the vicinity of the Pärvie Fault challenges the potential earthquake-induced origin. It is predominantly the scattering of a group of landslides across a discrete area, in close proximity to a fault, and their synchronous age rendering it likely that they were triggered by an earthquake (e.g., Jibson, 1996; Ojala et al., 2019). The spatial distribution of the two RSF deposits and the corresponding ages are therefore not enough evidence to conclude whether they were triggered by an earthquake or by other triggers, such as glacier debuttressing after deglaciation. However, the absence of a group of landslides could hint towards the nature of the Pärvie Fault rupture. It is in stark contrast to the large groups of earthquake-induced landslides nearby glacially-induced faults in northern Finland (e.g., Ojala et al., 2019). The presence of fault scarps but absence of landslides could support the

occurrence of earthquakes underneath the retreating ice sheet. The crosscut shorelines of Torneträsk indicate that the fault scarps locally ruptured at a close distance to the retreating ice margin. Although there is mounting evidence that the Pärvie Fault was not the result of a single rupture, it cannot be ruled out that there was a partial subglacial rupture. Sutinen et al. (2019) suggests morphological signs of subglacial rupture could be anastomosing networks of eskers (Fig. 10b) and subglacial crevasse fillings, which are both present in the Torneträsk area (Ploeg, 2022)."

**Other relevant changes**

- *Lines 62-65: "Refining ice-dammed lake reconstructions impacts the precision of reconstructed patterns of ice sheet retreat (e.g., Jansson, 2003; Utting and Atkinson, 2019; Regnéll et al., 2019, 2023; Dulfer, et al., 2022), which is especially valuable as the dynamics of ice sheet demise in topographically challenging terrain remains understudied in Scandinavia (Borgström, 1989; Kleman et al., 2020; Regnéll et al., 2019, 2023)."*

  We added a reference to a recent Norwegian study using ice-dammed lakes for reconstructing retreat patterns in mountainous regions (Romundset et al., 2023), whose omission was an oversight on our behalf.

  Lines 62-66:
  "Refining ice-dammed lake reconstructions impacts the precision of reconstructed patterns of ice sheet retreat (e.g., Jansson, 2003; Utting and Atkinson, 2019; Regnéll et al., 2019, 2023; Dulfer, et al., 2022; Romundset et al., 2023), which is especially valuable as the dynamics of ice sheet demise in topographically challenging terrain remains understudied in Scandinavia (Borgström, 1989; Kleman et al., 2020; Regnéll et al., 2019, 2023; Romundset et al., 2023)."

- *Lines 383-385: "The calculations strongly depend on the direction of the tilt along which they were calculated, the resolution and accuracy of the DEM, the precision of the mapping of shorelines, and on post-depositional faulting."*

  We have removed mention of "the resolution and accuracy of the DEM", as we do not consider this to be a factor of uncertainty due to the high resolution. The word "strongly" is also removed, as the differences in gradient as a result of the above-mentioned factors is actually minimal.

  Lines 498-499:
  "The calculations depend on the direction of the tilt along which they were calculated, the precision of the mapping of shorelines, and on post-depositional faulting."

**New references to the manuscript after reviewing**

Boyes, B. M., Linch, L. D., Pearce, D. M., and Nash, D. J.: The last Fennoscandian Ice Sheet glaciation on the Kola Peninsula and Russian Lapland (Part 2): Ice sheet margin positions, evolution, and dynamics, Quaternary Sci. Rev., 300, 107 872, https://doi.org/10.1016/j.quascirev.2022.107872, 2023.

Goodfellow, B. W., Stroeven, A. P., Lifton, N. A., Heyman, J., Lewerentz, A., Hippe, K., Näslund, J.-O., and Caffee, M. W.: Last ice sheet recession and landscape emergence above sea level in east-central Sweden, evaluated using in situ cosmogenic 14C from quartz, Geochronol., 6, 291–302, https://doi.org/10.5194/gchron-6-291-2024, 2024.705

Hughes, A. L., Clark, C. D., and Jordan, C. J.: Subglacial bedforms of the last British Ice Sheet, J. Maps, 6, 543–563, https://doi.org/10.4113/jom.2010.1111, 2010.

Jibson, R. W.: Use of landslides for paleoseismic analysis, Eng. Geol., 43, 291–323, https://doi.org/10.1016/S0013-7952(96)00039-7, 1996.

Lantmäteriet: Product description: Orthophoto (Document version 2.6), https://www.lantmateriet.se/globalassets/geodata/geodataprodukter/ flyg--och-satellitbilder/e_pb_ortofoto.pdf, accessed 8 November 2024, 2021a.

Ojala, A. E. K., Mattila, J., Markovaara-Koivisto, M., Ruskeeniemi, T., Palmu, J.-P., and Sutinen, R.: Distribution and morphology of landslides in northern Finland: An analysis of postglacial seismic activity, Geomorphology, 326, 190–201, https://doi.org/10.1016/j.geomorph.2017.08.045, 2019.

Ploeg, K.: Glacial lakes in the Torneträsk region, northern Sweden, are key to understanding regional deglaciation patterns and dynamics, Master's thesis, Stockholm University, Stockholm, http://urn.kb.se/resolve?urn=urn:nbn:se:su:diva-208142, 2022.

Romundset, A., Akçar, N., Fredin, O., Andersen, J. L., Høgaas, F., Christl, M., Yesilyurt, S., and Schlüchter, C.: Early Holocene thinning and final demise of the Scandinavian Ice Sheet across the main drainage divide of southern Norway, Quaternary Sci. Rev., 317, 108 274, https://doi.org/10.1016/j.quascirev.2023.108274, 2023.